# MEMORY MOSAICS

**Jianyu Zhang**[†1]**, Niklas Nolte**[†]**, Ranajoy Sadhukhan**[‡]**, Beidi Chen**[†‡]**, Léon Bottou**[†1]
[†] FAIR, Meta    [‡] Carnegie Mellon University    [1] New York University

## ABSTRACT

Memory Mosaics are networks of associative memories working in concert to achieve a prediction task of interest. Like transformers, memory mosaics possess compositional capabilities and in-context learning capabilities. Unlike transformers, memory mosaics achieve these capabilities in comparatively transparent way ("predictive disentanglement"). We illustrate these capabilities on a toy example and also show that memory mosaics perform as well or better than transformers on medium-scale language modeling tasks.

## 1 Introduction

This paper presents a learning system architecture, *Memory Mosaics*, in which multiple associative memories work in concert to carry out a prediction task of interest. Such systems are closely related to memory networks (Weston et al., 2014; Sukhbaatar et al., 2015) and resemble transformers (Vaswani et al., 2017) despite significant differences. Like transformers, Memory Mosaics possesses some of the disentanglement and compositional capabilities that have long eluded machine learning systems (Lake & Baroni, 2018). Unlike transformers whose internal mechanism are hard to decipher (Olsson et al., 2022; Bietti et al., 2024), Memory Mosaics achieve these capabilities in comparatively transparent ways.

The three main contributions of this work are (a) defining an architecture that exploits the direct similarity between self-attention and associative memories implemented with kernel regression, (b) identifying and illustrating the predictive disentanglement principle which explains how training decomposes the overall task in interesting ways, and (c) showing that this comparatively transparent architecture matches the i.i.d. performance of decoding transformers on a language modeling task, and outperforms them on o.o.d. tasks such as in-context learning.

Section 2 reviews related work. Section 3 describes simple associative memory units than can be inserted in a deep network. Section 4 explains how training such a network splits a prediction task into disentangled sub-tasks. Section 5 illustrates this "predictive disentanglement" using a network with only 54 parameters, showing that this is not a mysterious effect of scale but a property of the architecture. Section 6 extends these ideas to fully formed memory mosaics. Section 7 reports on medium-scale language modeling experiments.

## 2 Related Work

Several recent papers (e.g., Katharopoulos et al., 2020; Peng et al., 2023; Sun et al., 2023; Gu & Dao, 2023) propose transformer alternatives that use efficient recurrences to cut the quadratic computational cost of transformers. Closer to our interests, other authors (e.g., Ramsauer et al., 2020; Krotov, 2023; Hoover et al., 2024) rethink transformers with Hopfield-style associative memories and their associated energy function. In contrast, we leverage elementary associative memories that interpolate stored key/value pairs with a kernel regression (therefore incurring a quadratic runtime cost) in order to construct an architecture that remains very close to standard transformers but cast a new light on properties that play an important role in their compositional learning capabilities.

Closely related to predictive disentanglement, (Bengio et al., 2019) proposes a meta-learning training objective that achieves causal disentanglement by seeking quick adaptation to new distributions. We argue that a similar effect happens in our architecture, as a consequence of the normal training process interpreted as a meta-learning process, revealing an important aspect of the still mysterious compositional learning abilities of transformer-like architectures.

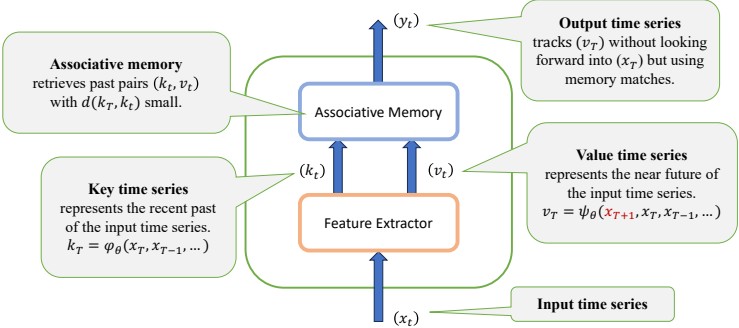

Figure 1: Elementary memory unit. The keys $k_T$ are computed as a function of past observations $(x_t)_{t \leq T}$. The values $v_T$ peek into the future. In this example, the value also depend on the next observation $x_{T+1}$. At time $T$, the associative memory uses the known key $k_T$ to compute an estimate $y_T$ of $\mathbb{E}(v_T | k_T)$ using only the previously stored pairs $(k_t, v_t)$, $t < T$. One time step later, the input $x_{T+1}$ is revealed, the value $v_T$ can be computed, and the pair $(k_T, v_T)$ is added to the memory.

## 3  Memories

**Associative memory**   Generally speaking, an associative memory is a device that can store key-value pairs and retrieve values given a corresponding key. This definition omits important details about dealing with duplicate keys and approximate matches. For our purposes, both keys and values shall be vectors in $\mathbb{R}^d$. The retrieval process can then be represented as a function of the queried key $k$ and all the stored pairs $(k_1, v_1) \ldots (k_n, v_n)$.

$$
\begin{cases}
\mathbb{R}^d & \to & \mathbb{R}^d \\
k & \mapsto & f\big(k;\ \{(k_1, v_1) \ldots (k_n, v_n)\}\big)
\end{cases}
$$

Except perhaps when duplicate keys are involved, an associative memory stores key-value pairs without consideration for their temporal ordering. Therefore the retrieval function can be assumed invariant with respect to any permutation of the stored pairs. This exchangeability property suggests that we can also view an associative memory as a device that estimates a conditional probability distribution $P(V|K)$ on the basis of the sample $(k_1, v_1) \ldots (k_n, v_n)$ of key-value pairs. The retrieval function is then a conditional expectation over this estimated distribution:

$$
f\big(k;\ \{(k_1, v_1) \ldots (k_n, v_n)\}\big)\ =\ \mathbb{E}(V \mid K = k). \tag{1}
$$

Such a conditional expectation can be constructed with Gaussian kernel regression,[1]

$$
f\big(k;\ \{(k_1, v_1) \ldots (k_n, v_n)\}\big)\ =\ \sum_{i=1}^{n} \frac{1}{Z}\, e^{-\beta \|k - k_i\|^2} v_i \quad \text{with} \quad Z = \sum_{i=1}^{n} e^{-\beta \|k - k_i\|^2}. \tag{2}
$$

The close connection between this Gaussian kernel smoothing and attention (Bahdanau et al., 2015) is obvious when all key vectors $k_i$ share a same squared norm because expression (2) becomes

$$
f\big(k;\ \{(k_1, v_1) \ldots (k_n, v_n)\}\big)\ =\ \sum_{i=1}^{n} \frac{e^{\beta\, k^\top k_i}}{\sum_{j=1}^{n} e^{\beta\, k^\top k_j}}\, v_i. \tag{3}
$$

There are of course more advantageous ways to implement associative memories. Although some will certainly prove useful in the future, this paper only relies on associative memories implemented with Gaussian kernel smoothing, not least because that makes it easy to compute gradients.

**Predicting with associative memories**   Consider now a sequence $(x_t)$ of observations, discrete tokens or continuous values. We would like to leverage the past observations $(x_t)_{t \leq T}$ to predict some useful property of the future observations $(x_t)_{t > T}$. For instance we might want to predict the next observation $x_{T+1}$ to construct an auto-regressive model of the sequence.

---

[1]Expression (2) is known as the Nadaraya-Watson estimator (Nadaraya, 1964; Watson, 1964). It is known to converge to the true conditional expectation $\mathbb{E}(K|V)$ when $n \to \infty$ and $\beta = \sqrt{n}$.

Our elementary memory unit (Figure 1) consists of an associative memory and a trainable feature extractor that computes suitable keys and values for the memory. The keys $k_T$ are computed as a function of the past observations $(x_t)_{t \leq T}$ and trainable weights $\mathbf{w}$,

$$k_T = \varphi(x_T, x_{T-1}, \dots; \mathbf{w}) \,. \tag{4}$$

In contrast, the values $v_T$ are allowed to peek in the future because they represent what the memory module aims to predict. For instance, the systems described in this paper merely allow values to depend on the next observation $x_{T+1}$,

$$v_T = \psi(\mathbf{x_{T+1}}, x_T, x_{T-1}, \dots; \mathbf{w}) \,. \tag{5}$$

The memory units operate *independently* at *inference time*. They start empty at the beginning of each input sequence. At time step $T$, each memory receives a key vector $k_T$ computed from the recent inputs $(x_T, x_{T-1}, \dots)$ and interpolates a response $y_t$ on the basis of the previously stored key/value pairs. The value $v_T$ is computed one time step later when the next input $x_{T+1}$ is revealed and the pair $(k_T, v_T)$ is added to the memory.

Although the value $v_T$ depends on the near future, the output $y_T$ does not depend on $v_T$ but merely leverages the previously stored key/value pairs to estimate $v_T$. Therefore there is no leak of future information: each memory unit is a little machine that predicts a bit of future information (described by $v_T$) on the basis of recent information (described by $k_T$) and previously stored key/values pairs.

The exact form of the feature extraction functions can vary in complexity. For instance, when each observation $x_T$ carries sufficient information, the keys $k_T$ and values $v_T$ can be computed as linear functions of respectively $x_T$ and $x_{T+1}$, that is $k_T = W_\varphi x_T$ and $v_T = W_\psi x_{T+1}$. However we find useful to consider feature extraction functions that summarize the *recent past* using short convolutions or quickly vanishing leaky averages. For instance, the language experiments of Section 7 use feature extractors of the following form:[2]

$$
\begin{array}{lll}
k_T = \mathrm{Norm}(\bar{k}_T) & \text{with} & \overbrace{\bar{k}_T = \tilde{k}_T + \lambda_\varphi \bar{k}_{T-1}}^{\text{leaky average over } t = T, T-1\dots, 1} \quad \tilde{k}_T = W_\varphi x_T \\
v_T = \mathrm{Norm}(\bar{v}_T) & \text{with} & \underbrace{\bar{v}_T = \tilde{v}_T + \lambda_\psi \tilde{v}_{T+1}}_{\text{convolution over } t=T \text{ and } T+1} \quad \tilde{v}_T = W_\psi x_T
\end{array} \tag{6}
$$

Since this expression produces keys with unit norm ($\mathrm{Norm}(x) = x/\|x\|$), the effective kernel bandwidth is determined by the trainable parameter $\beta$ in equation (3).

**Training networks of memory units** Consider now a deep network whose architecture includes layers of associative memory units. When the associative memories are implemented with differentiable kernel smoothing mechanisms, training such a deep network is simply a matter of unrolling the network in time and back-propagating the gradients, in ways that users of modern deep learning software will find very familiar. Unsurprisingly, unrolling equation (3) along an input sequence $(x_1 \dots x_D)$ of duration $D$ yields an expression that very much resembles masked self-attention (Vaswani et al., 2017).

$$\forall T \in \{1 \dots D\} \qquad y_T = \sum_{i=1}^{T-1} \frac{e^{\beta\, k_T^\top k_i}}{\sum_{j=1}^{T-1} e^{\beta\, k_T^\top k_j}}\, v_i \,, \tag{7}$$

Implementing associative memories with kernel smoothing therefore provides a particularly direct illustration of the connection between self-attention and associative memories (e.g., (Ramsauer et al., 2020)). However, Memory Mosaics differ because the value extraction function is allowed to peek into the near future of the input time series $(x_t)$. This slight change has important consequences

- Each memory unit operates as a little predictor whose outputs $y_T$ can be interpreted as a conditional expectation (1) that estimates features of the near future ($v_T$) of the input time series on the basis of its past observations ($k_T$). The parameters of the value extraction function ($\psi$) specify what is being predicted and the parameters of the key extraction function ($\varphi$) specify how it is predicted.

---

[2]The leaking average in expression (6) is far too simple to effectively encode long range dependencies as demonstrated in (Voelker et al., 2019; Peng et al., 2023; Gu & Dao, 2023).

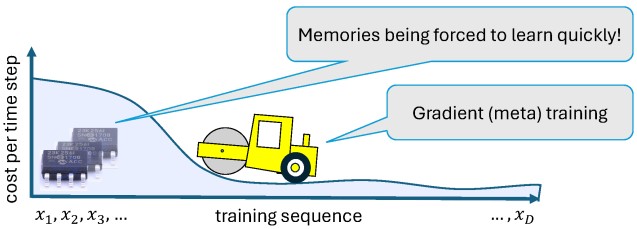

Figure 2: The curve plots the prediction losses for all training sequence indices $t \in \{1 \ldots D\}$ in the training sequence. Minimizing their sum —the area under the curve— favors memories that produce useful value estimates after fewer time steps.

- Equation (7) must therefore account for the number of future time steps needed to compute $v_T$. In our experiments, for example, $v_T$ can look one step ahead in the future. This amounts to having a more aggressive attention mask. Therefore the main diagonal must be excluded from the attention mask, justifying the $T-1$ upper bound in the sum.[3]

- Because each memory unit acts as a predictor, a single layer of memory units is sufficient to address the induction head problem of Bietti et al. (2024). In contrast, a decoding transformer needs at least two self-attention layers for the same task.

- Equation (7) makes no provision for position encoding and no distinction between query and key vectors. In other words, we are betting that these transformers complications are no longer needed because our associative memory units do not need them to implement induction heads.

## 4 Predictive Disentanglement

**Training and meta-learning** The training process determines which future bit of information is predicted by each associative memory unit (through the parameters that control the computation of the values $v_T$) and which kernels are used to perform the predictions (through the parameters of that control the computation of the keys $k_T$). In contrast, *the relation between keys and predicted values is determined for each input sequence at inference time* through the memorization of key/values pairs specific to each sequence. The training procedure should therefore be seen as a *meta-learning process*, distinct from the memory-based learning that occurs at inference time when new key/value pairs are added into the memories.

**Predictive disentanglement** This meta-learning interpretation reveals a remarkable phenomenon that we call *predictive disentanglement* : the gradient training algorithm splits the overall prediction task (e.g., predicting the next token in a natural language sentence) into disentangled prediction sub-tasks assigned to each memory unit.

Consider a training set composed of long enough sequences $(x_1, \ldots x_D)$ extracted from underlying time series governed by possibly different stationary processes. The goal of our network is to predict each $x_{T+1}$ using the previous observations $x_1 \ldots x_T$. Unrolling the network in time along each sequence $(x_1 \ldots x_D)$ and collecting the prediction losses measured at each position $t$ can be summarized by a curve that shows the prediction cost (or loss) at each time step $1 \ldots D$, as illustrated in Figure 2. We can expect that the prediction cost observed at position $T$ becomes smaller when $T$ increases because more information $(x_1 \ldots x_T)$ is available to predict each $x_{T+1}$.

The training process minimizes the total prediction cost, that is the area under the curve in Figure 2 viewed as a collection of vertical slices. We can also view this area as a collection of horizontal slices, each representing the context length required to drive the prediction cost below a certain threshold. Therefore the training process can also be viewed as *minimizing the context length needed to produce good enough predictions*.

Because the associative memory retrieval function (2) is known to converge to stationary conditional expectations $\mathbb{E}(V|K)$, each memory unit is driven to produce a good conditional expectation estimate as soon as possible. This can be achieved in two ways:

- Let us first assume that each memory unit has a frozen value extraction function $\psi$. The training procedure can still make each memory unit statistically more efficient by tuning the parameters

---

[3]One could of course use a more aggressive masking to allow $v_T$ peeking several time steps in the future.

of the key extraction function $\varphi$, that is, by learning how to compare the current prediction context $(x_T, x_{T-1}, x_{T-2} \dots)$ with past prediction contexts $(x_t, x_{t-1}, x_{t-2} \dots)$ for $t < T$.

Learning a similarity metric (a kernel) is a well known way to make non-parametric estimators more efficient (e.g., Bach et al., 2004). For instance, the training procedure can construct keys that summarize the relevant contextual information, discarding noise factors that could increase the distance between keys associated with similar values. It can also adjust the effective kernel bandwidth, for instance, using parameter $\beta$ in equation (7).

- When multiple memory units are available, the training procedure can also *distribute* the overall prediction task among the available memory units. As long as the memory units outputs can still be combined to address the overall task, the training algorithm can optimize the parameters of the value extraction functions $\psi$ to produce values $v_T$ that more efficiently modeled by their respective memory units.

  Because each memory unit operates independently at inference time, this works best when the overall prediction task is *disentangled into smaller prediction sub-tasks that can be modeled independently and efficiently*. More precisely, the sub-tasks must be chosen so that each memory can carry out its assigned modeling task at inference time without having to account for the combined impact of the operation of all memory units. Their outputs can then be recombined to provide predictions for inputs that are globally very different from the training inputs, but whose disentangled components are individually predictable, as illustrated in Section 5.

Disentanglement has long been recognized as desirable (Bengio, 2013) but has been hard to pinpoint (Comon, 1994; Roth et al., 2022; Thomas et al., 2018). Predictive disentanglement is closely related to the meta-transfer objective of Bengio et al. (2019) but arises as a side effect of a specific predictive architecture trained with the usual gradient procedure. Although predictive disentanglement is easier to understand in the case of a network of associative memory units, we conjecture that something similar also occurs in standard transformers.

## 5 Tracking three moons

We give an illustrative example of predictive disentanglement: three moons orbit a remote planet. Although the local astronomers are very far from understanding celestial mechanics,[4] they nevertheless observe periodic motions and debate how to predict future moon positions. A first astronomer proposes to compile a single table containing the daily positions of all three moons, arguing that if the current set of moon positions matches a previous observation, the future moon positions will match the following observations. A second astronomer suggests instead to make three tables, one for each moon, arguing that the future positions of each moon can be independently predicted by matching its current position with a previously observed one.

To make reliable predictions, the first astronomer needs a table that contains at least one record for each of the possible moon configurations. Our astronomer therefore needs to log the daily moon positions until all three moons return to their original configuration, after a number of days equal to the least common multiple $\mathrm{lcm}(p_1, p_2, p_3)$ of the individual moon periods. In contrast, the second astronomer only needs to log daily moon positions until each of the moons returns to a previously observed position, for a number of days equal to the period $\max(p_1, p_2, p_3)$ of the slowest moon.

One could argue that the proposal of the second astronomer is obviously superior because the three moons are distinct objects, well separated in space and time. One could instead argue that we view the moons as separate objects precisely because their respective futures can in general be independently predicted. Space and time separation merely suggests the possibility of independent predictions, as long as the moons do not collide.

**Model**  For our purposes, each observation $x_t$ consists of three complex numbers $e^{i\theta_k}$ that encode the angular positions $\theta_k$ of the three moons inside their respective orbital plane. We consider two single layer models (Figure 3) with either $N_h = 1$ or $N_h = 3$ memory units whose added dimensions match the input dimension. The trainable parameters of the linear key and value extraction are collected in two $3 \times 3$ complex matrices $W_\varphi$ and $W_\psi$. The memory unit follow equation (3) with a fixed parameter $\beta = 50$. A third $3 \times 3$ complex matrix $W_z$ combines the memory unit predictions into an output $z_T$ that hopefully predicts $x_{T+1}$. Both networks share an interesting analytic solu-

---

[4]We do not seek to discuss subtleties such as elliptical orbits or multi-body problems. Our primitive astronomers are best compared to the ancient sky watchers whose efforts eventually gave the Ptolemaic model.

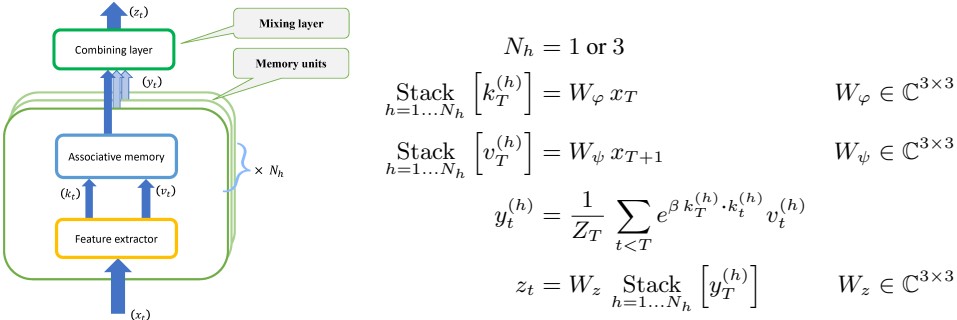

Figure 3: An architecture for the three moons problem. We consider single-layer networks with either $N_h = 1$ or $N_h = 3$ memory units whose keys and values belong to either $\mathbb{C}^3$ ($N_h = 1$) or $\mathbb{C}^1$ ($N_h = 3$). Both nets have $3 \times 3 \times 2 \times 3 = 54$ trainable real parameters that determine how to predict the moon positions using either a single 6-dimensional memory or three 2-dimensional memories.

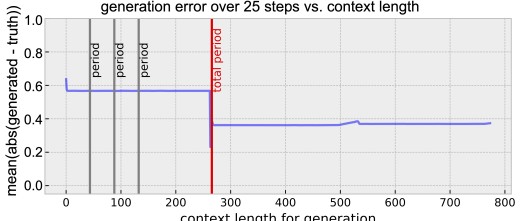

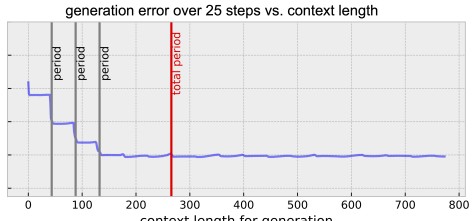

Figure 4: Single head network prediction error versus context length. The prediction error shows a sharp transition after $\text{lcm}(p_1, p_2, p_3)$ observations (red vertical line), when the network switches from predicting the future moon position by repeating the last observation to predicting by find a matching memorized configuration.

Figure 5: Three-heads network prediction error versus context length. The prediction error improves whenever the context length reaches the period of a new moon (black vertical lines), yielding accurate predictions after the last one, well before having seen the full set of moon configurations (red vertical line).

tion: setting all three matrices $W_\varphi$, $W_\psi$, and $W_z$ to the identity yields optimal predictions once the associative memories have seen enough samples.

**Training** The networks are trained using randomly generated sequences $(x_t)$ of length 800. Each sequence features three moons whose periods are related by randomly chosen ratios and are scaled to ensure that the 800 observation sequence contains at least three full periods $\text{lcm}(p1, p2, p3)$ of the moon system. Validation sequences are constructed similarly using a set of moon periods that does not appear in the training set.

Figure 4 and 5 show the prediction errors of both networks as a function of the context length, that is, the number of observations stored into the memories. More precisely, for each sequence $(x_t)$ and each time index $T$, we compute the average absolute deviation between the next 25 true moon positions $x_{T+1} \ldots x_{T+25}$ and the next 25 auto-regressive predictions (in which the successive predictions are looped back into the network input.) The plots show curves averaged over 512 sequences sharing the same set of moon periods taken from either the training or validation set.

- For the single head network (Figure 4), the plots show a sharp transition after $\text{lcm}(p_1, p_2, p_3)$ observations, that is, when the memory contains a full set of moon configurations (red vertical line). Before this threshold, predictions are performed by repeating the last observation. After this threshold, predictions are performed by finding a matching moon configuration in the memory, just as suggested by the first astronomer.

- For the three-heads network (Figure 5), the prediction error curve drops after seeing exactly $p_1$, $p_2$, and $p_3$ observations (black lines), that is whenever the orbit of an additional moon has been memorized. The learned weight matrices are shown Figure 10 in the Appendix. Observe how the network produces accurate predictions after a time equal to the period $\max(p_1, p_2, p_3)$ of the slowest moon (last black line), long before the combined period $\text{lcm}(p_1, p_2, p_3)$ (red line) of the

moon system. In this interval, *accurate predictions are returned for moon configurations that can be very different from the previously observed ones.* Instead the network *combines individual moon predictions, each well supported by the past observations.*

**Predictive disentanglement and compositional learning in language models**   Consider a chat-bot assisted creative writing scenario in which the human uses dialogue to repeatedly introduce new ideas into an evolving story that the chat-bot reprints at each step. The user can drive such a story arbitrarily far from the training data and into the distant tail of its distribution. Although no training example resembles the story, the chat-bot keeps producing syntactically correct language and coherent stories because it has learned some of the mathematical structures of language (Harris, 1968) and can recombine pieces of information coming from either the context or the training data. This phenomenon is fundamentally similar to that illustrated in Figure 5, where moon configurations unlike any previously seen configurations are accurately predicted because the network has learned how to combine individual moon predictions. This similarity casts a useful light on the otherwise mysterious compositional learning abilities of transformer-like models.

## 6   Layered memories

We of course envision deeper networks of memory units. In order to make meaningful comparisons, we also would like to remain as close as possible to the classic transformer architecture which alternates self-attention layers with fully connected feed-forward networks (FFNs).

**Persistent memories**   Sukhbaatar et al. (2019) shows that FFNs in a transformer can be interpreted as *persistent memories* that augment the self-attention layers and provide means to represent information that persists across input sequences. Besides the *contextual memory units* (Figure 1), we therefore introduce *persistent memory units* (Figure 12 in the Appendix) that contain a predefined number of key value pairs $(k_i, v_i)_{i=1...N_m}$ determined at training time through gradient backpropagation. Persistent memory units no longer need an explicit value extraction function because the memory content is not updated at inference time. As pointed out by Sukhbaatar et al., they also can be viewed as fully connected neural networks with a single hidden layer that uses a soft-max non-linearity instead of a component-wise transfer function. Yet, we find conceptually useful to still view the persistent memory output $y_t$ as the conditional expectation $\mathbb{E}(V|K)$ of an implicit value function that is not explicitly parameterized, but can be figured out after training.

**Routing**   Interleaving layers of contextual and persistent memory units can then be understood as means to increase the effective complexity of either the feature extractors or the combining layers of contextual memories (see Figure 6 for a spoiler). Therefore persistent memory units can also be seen as tool for *routing information* between successive layers of contextual memory units. Such a circuitry can implement routes that depend on the data, just like the gating modules of a mixture of expert (Jacobs et al., 1991). Since all the parameters of such a circuitry are determined at training time, all the possible routes would have to be determined at training time. However the learning algorithm can overcome this limitation by also recruiting contextual memory units from adjacent layersy. Because the contents of contextual memory units are updated at inference time, recruiting some of them into the routing circuitry provides the means to create new routes on the basis of the first observations of a new sequence, suggesting an efficient alternative to capsule networks (Sabour et al., 2017).

**Memory Mosaics**   In such a complex network, the division of labor between contextual memory units is still determined by the predictive disentanglement principle. During training, the steamroller of Figure 2 pushes the contextual memory units towards functions that more easily memorized independently than in aggregation. This does not only hold for memory units that record primary pieces of information such as the moon positions of Section 5, but also for those that affect the routing circuitry and those that operate on the information produced by earlier memory units.

Therefore, *under the pressure of the predictive disentanglement principle*, a network of memory units does *not only memorize disentangled fragments of information, but also memorizes how they fit together and how their combinations can be again broken into new disentangled fragments and recombined in myriad ways.* This is why we call such networks *Memory Mosaics*.

## 7   Modeling language with memories

We have so far described Memory Mosaics as an architecture that resembles transformers in important way but offers additional insights such as predictive disentanglement. We now provide evidence

that Memory Mosaics can handle the most successful application of decoding transformers, that is, language modeling.

**Language modeling task**   The TINYSTORIES work of Eldan & Li (2023) shows how to study large language modeling questions using small language models. This is achieved by limiting the scope to tiny stories written in simple english and taking place in the simple world that a three years old child could understand. A small language model trained on such data generates continuations with far better language quality and narrative consistency than those a much larger model (1.5B parameters) trained on a generic text.

Following both the lead of Eldan & Li and the advice of our legal department, we leverage the Mixtral-8x7B open language model (Jiang et al., 2024) to generate a new corpus of tiny stories dubbed BABISTORIES. This corpus and its generation are detailed in Appendix B.[5]

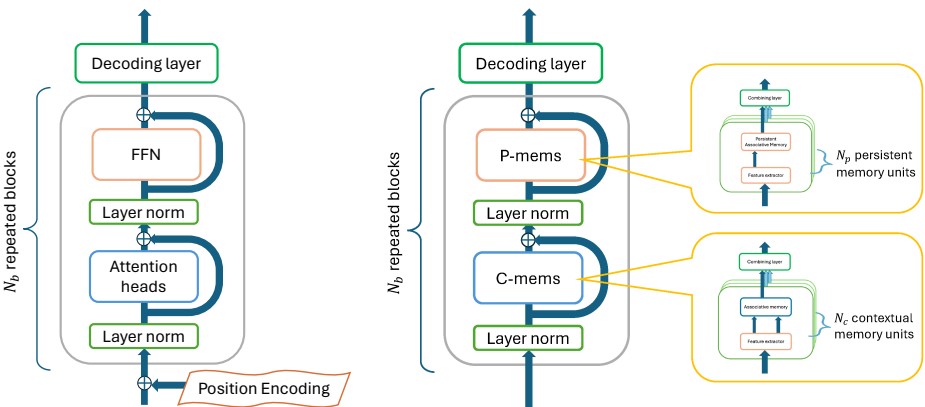

Figure 6: *Left*: Classic GPT2-small transformer. *Right*: GPT2-like Memory Mosaic

**Architecture**   To put our experiments into context, we design a Memory Mosaic architecture that closely matches the classic GPT2-small transformer architecture (Radford et al., 2018; 2019). Both architectures, shown side-by-side in Figure 6, use the same GPT2 tokenizer, the same embedding dimension ($d = 768$), and the same number of heads ($N_h = N_c = N_p = 12$). Both architectures are trained and tested using sequences of length 512, that is, one to three stories long.

There are three major differences between these two architectures. First, the Memory Mosaic does not use positional encoding. Second, unlike the $N_h = 12$ attention heads of each transformer block, the $N_c = 12$ contextual memory units in each block do not distinguish keys from queries (Figure 1) but instead use the key and value extraction functions described in Equation 6. The keys are formed with a leaky average of past inputs, and the values can peek one time step ahead.[6] Accordingly, the attention mask excludes the main diagonal to avoid breaking causality. Finally, the feed forward networks (FFNs) of the classic transformers blocks are replaced by a layer of $N_p = 12$ persistent memory units, complete with a key extraction functions (6) and combining layer. These persistent memory units are sized to ensure that the per-block parameter count of the Memory Mosaic architecture closely matches GPT2-small.[7]

**Training and validation**   Figure 7 shows the training and validation curves of both transformers and Memory Mosaics of different depth trained on BABISTORIES. The Memory Mosaic slightly outperforms the transformer for small depth networks,[8] but this effect disappears when the depth increases and both the training and validation losses become indistinguishable. Additional results are presented in Appendix D.2.

---

[5]We share the BABISTORIES dataset and Memory Mosaics source code at `https://github.com/facebookresearch/MemoryMosaics`.

[6]The key idea here is to define key and value extraction functions that combine a couple successive inputs $x_t$ instead of just one as in the three moons example. Many variations perform more or less equivalently.

[7]Compared with GPT2-small, we save $768 \times 512$ position encoding weights and $N_b \times 768^2$ query projection weights, but add $2 \times N_b \times 768^2$ weights for the persistent memory key extraction and mixing layer. The total number of persistent memory unit slots is therefore close to the total number of FFN hidden units.

[8]This is not surprising because Memory Mosaics only need a single block to implement induction heads, whereas transformers need at least two for the same task.

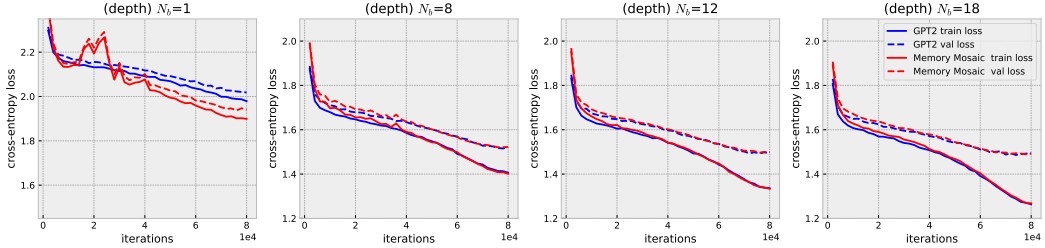

Figure 7: Training and validation loss of the transformer and Memory Mosaic architectures trained on BABISTORIES for different model depths. The horizontal axis represents the number of training iterations. All hyper-parameters have been tuned on the transformer architecture and transferred verbatim to the Memory Mosaic architecture. The Memory Mosaic slightly outperforms the transformer for small depth networks, but that effect disappears when the depth increases. Additional results are presented in Appendix D.2.

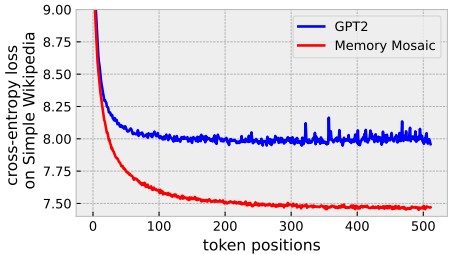

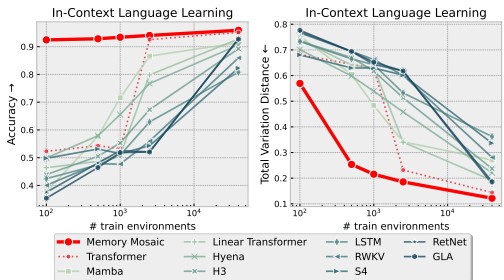

Figure 8: Prediction performance on the Simple English Wikipedia dataset using models trained on BABISTORIES. The plot shows the per-token average loss as a function of the position of the generated token in the 512-token long input window. Memory Mosaics outperform transformers after about 50 tokens, suggesting superior in-context learning abilities.

Figure 9: Memory Mosaics performance on the REGBENCH in-context learning benchmark (Akyürek et al., 2024). Since REGBENCH includes an hyper-parameter search, Memory Mosaics and transformers use the same search space with the same parameter counts. Memory mosaics outperform all previously tested architectures in this benchmark.

Importantly, all hyper-parameters were tuned for the transformer architectures (Appendix C) and transferred verbatim to the Memory Mosaics. This choice might explain why the training curves track each other so well. It also leaves the Memory Mosaics at a slight disadvantage.

**Qualitative evaluation** In order to compare the quality of the text generated by models trained on tiny stories, Eldan & Li designed twenty-four prompts that exercise the factual, logical, and consistency properties of the generated continuations. Table 4 in the Appendix compares the continuation generated on these prompts by a transformer and a Memory Mosaic, both $N_b = 18$ blocks deep. Both models perform very similarly on this task.

**Out-of-distribution evaluation** The Simple English Wikipedia[9] is a version of Wikipedia written in a language that is easier to understand. Despite the intended simplicity, the articles are substantially longer and more sophisticated than our BABISTORIES. Predicting Simple English Wikipedia articles using models trained on BABISTORIESis therefore a challenging out-of-distribution task.

Figure 8 shows the per-token average loss as a function of the position of the generated token in the input window. Both the transformer and the Memory Mosaic are $N_b = 12$ blocks deep. In this experiment, the token prediction is expected to improve when the increasing context size reveals that the distribution is different. The transformer performance plateaus after 100 to 150 tokens, which is

---

[9]Described in https://simple.wikipedia.org/wiki/Simple_English_Wikipedia with downloads in https://huggingface.co/datasets/wikipedia#20220301simple.

a bit shorter than a typical tiny story. Memory Mosaics substantially outperform transformers after about 50 tokens, suggesting superior in-context learning abilities.

**In-context learning evaluation**    In order to rigorously compare the in-context learning abilities of various architectures, the REGBENCH benchmark (Akyürek et al., 2024) constructs random artificial languages defined by probabilistic finite automata (PFA). Each input sequence is composed of 10 to 20 strings drawn from a same PFA and delimited separator tokens. The competing architectures are trained on a variable number of input sequences, then evaluated on their ability to predict the last token of testing sequences generated using held out PFAs.

Since REGBENCH performs a hyper-parameter searches, we use the Memory Mosaic architecture of Figure 6 with the same search space as transformers, ensuring that both transformers and Memory Mosaics have the same parameter count for the same architectural hyper-parameters. We sweep over depth $N_b \in \{2, 4, 8\}$, number of heads $N_h = N_c = N_p \in \{2, 4, 8\}$, embedding dimension in $d \in \{64, 128, 256\}$, weight decay in $\{10^{-2}, 10^{-1}\}$, and training epochs in $\{1, 2, \ldots 200\}$.

Figure 9 compares Memory Mosaic on REGBENCH with the results previously reported by Akyürek et al.. The left plot shows the prediction accuracy for the test string last token. The right plot compares the predicted last token distribution with the exact distribution implied by PFA. Memory Mosaics dominate this benchmark, substantially outperforming transformers, recurrent neural networks, and state-space models for training set sizes covering three orders of magnitude.[10]

# 8    Discussion

The starting point of this work is made of two very old ideas. The first one is augment a deep network with explicit memories. The second one is to let the learning process decide what gets memorized and how it gets retrieved. Although such ideas have been explored in memory networks (Weston et al., 2014; Joulin & Mikolov, 2015; Sukhbaatar et al., 2015), the importance of having lots of independent memories had not been fully appreciated.

This contribution focuses on networks of associative memories implemented with kernel smoothing, therefore amenable to gradient-based learning algorithms. Such learning machines not only resemble decoding transformers (Section 3) but also perform very much like decoding transformers on the sort of language modeling task that made them famous (Section 7). Although much work is needed to replicate our observations at far greater scale, Memory Mosaics satisfy narrative constraints as well as transformers (Table 4), and generally behave in very encouraging ways (Figures 8 to 15). In large-scale models, Zhang (2025) scale-up Memory Mosaics to 10B parameters, observing superior learning capabilities compared to Transformers (similar to the evaluations in Figures 8 and 9).

Most importantly, we understand what Memory Mosaics do far better than we understand what transformers do. First, the value extraction functions of the associative memory units precisely describe what each memory seeks to memorize. Second, the predictive disentanglement principle explains why training a Memory Mosaic breaks the overall prediction task into pieces that are more efficiently memorized when they are considered independently (Section 5). Therefore, Memory Mosaics are not just a transformer-like architecture, but also a model[11] for compositional learning systems that break knowledge into independently memorized fragments, then reassemble them as needed using combination strategies that can themselves be viewed as memorized knowledge fragments (Section 6).

The focus on memorization allow us to formulate new questions. Could memories operate independently on different time scales? Could we envision a richer memory hierarchy than simply distinguishing persistent memories from contextual memories? Can intermediate memory tiers be trained like contextual memories, that is, without gradients? Can the persistent knowledge be then reduced to a compact high order bias?

Memory Mosaics also offer an array of engineering opportunities. Limited storage contextual memories could leverage least-recently used eviction schemes (e.g., Xiao et al., 2023), and associative memories could be implemented using a wide spectrum of techniques, either classical (e.g., Greengard & Strain, 1991; Spring & Shrivastava, 2017), or neural (e.g., Krotov, 2023), which could redefine the computing requirements of contemporary artificial intelligence systems.

---

[10]Although the baseline methods trained with small training sets (e.g. 100) perform poorly on the REGBENCH task, they perform very well when tested in-distribution (see Table 3 in the Appendix). Therefore they learned to model the training languages but did not acquire the ability to learn new languages in context.

[11]Not as in "statistical model" but as in "model used to describe and explain a phenomenon."

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

# Memory Mosaics– Appendix

## A  Tracking three moons

Figure 10 shows how the training process yields parameter matrices $W_\varphi$, $W_\psi$, and $W_z$, that dedicate one memory unit to each moon.

Training the three-heads network can be quite challenging in a manner that resembles the XOR networks of the early times (Rumelhart et al., 1986). We obtained reliable convergence using two tricks. First, we slightly restrict the linear operations by using $3 \times 3$ complex matrices (18 real parameters) instead of $6 \times 6$ real matrices (36 real parameters) operating on the 3-dimensional complex vectors as 6-dimensional real vectors. Second, we clip the mean squared loss in order to prevent the training algorithm from trying to optimize the prediction error when the memories are nearly empty.[12]

Reliable convergence could also be achieved by making any of $W_\varphi$, $W_\psi$, or $W_z$ equal to the identity. Doing so would of course bias the network toward the disentangled solution, something we wanted to avoid. Yet it is not unreasonable to believe that disentanglement can often be achieved in the canonical basis. For instance, objects well separated in space often appear in different image regions, and therefore along different pixels axes.

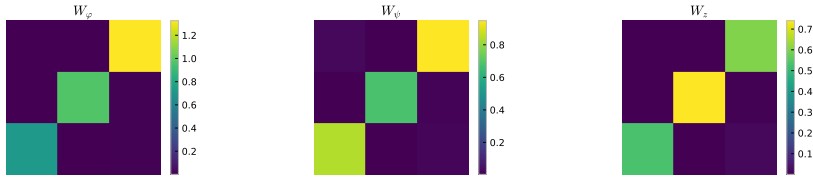

Figure 10: Visualization of the disentangled $W_\varphi$, $W_\psi$, and $W_z$ matrices in the 3-heads network. The color scale represents the moduli of the complex matrix coefficients.

---

[12]The steamroller metaphor (Figure 2) makes more sense when the loss is bounded.

# B  BabiStories

The TINYSTORIES dataset (Eldan & Li, 2023) is composed of stories written in a simple language and taking place a narrow world. Such stories can be used to train relatively small language models that still must address some of the broader language modeling challenges such as obeying narrative necessity and maintaining logical consistency. This dataset is a wonderful way to study big problems with acceptable computation and quick turn around.

The experiments of Section 7 were carried out using a dataset generated using a similar methodology but using the MIXTRAL-8X7B open language model in order to generate unencumbered data. We call this dataset BABISTORIES. All the scientific credit is still due to the remarkable work of Eldan & Li. Table 1 provides basic statistics for this newly generated BABISTORIESdataset, essentially matching those of the original TINYSTORIES dataset of Eldan & Li (2023). We had to increase the diversity of the generated stories by expanding the prompt to specify first names and by providing opening words for the story, in addition to required words and story features used by Eldan & Li (Figure 11). We also removed the few generated stories containing URLs.

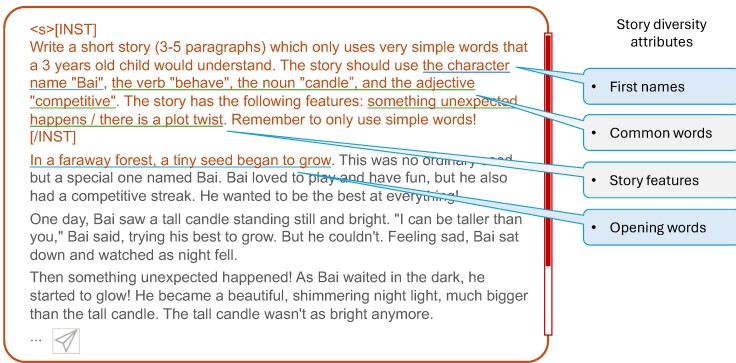

Figure 11: Generation of BABISTORIES. In order to improve the diversity of the generations, each story is generated by a prompt that provides a list of required words and story features (as in Eldan & Li, 2023) and additionally provides first names and opening words.

Table 1: BABISTORIES statistics.

| dataset partition | #stories | #tokens (GPT2 tokenizer) | #char per story (average) |
|---|---|---|---|
| train | 2.2M | 474,704,907 | 888 |
| valid | 2.2k | 4,749,107 | 889 |

## C GPT2 baseline and hyperparameters

Table 2 showcases the hyper-parameters searching process of GPT2 transformer baseline on the BABISTORIES dataset, where we use AdamW optimizer Loshchilov & Hutter (2017), batch-size 512, context-size 512, and a cosine learning rate scheduler with minimum learning rate $1e - 4$ for all training.

Table 2: Hyperparameters searching on GPT2 transformer with $N_b = 12$. "dropout", if any, is applied on attention score, attention heads output (before combining layer), and FFN output.

| learning rate | dropout | L2 weight decay | warm-up iters | training iters | train loss | valid loss |
|---------------|---------|-----------------|---------------|----------------|------------|------------|
| 5e-3 | 0.05 | 0.1 | 2000 | 80000 | 1.336 | **1.494** |
| 1e-3 | 0.05 | 0.1 | 2000 | 80000 | 1.350 | 1.524 |
| 5e-3 | 0 | 0.1 | 2000 | 80000 | 1.281 | 1.556 |
| 5e-3 | 0.05 | 0.01 | 2000 | 80000 | 1.322 | 1.516 |
| 5e-3 | 0.05 | 0.1 | 200 | 80000 | fail | fail |
| 5e-3 | 0.05 | 0.1 | 2000 | 40000 | 1.325 | 1.532 |
| 5e-3 | 0.05 | 0.1 | 2000 | 160000 | 1.314 | 1.497 |

# D  Memory Mosaics for language modeling

## D.1  PERSISTENT MEMORY UNITS

Persistent memory units produce their outputs using the same key extraction function $\varphi(x_T, x_{T-1}, \dots)$ and the same retrieval function (3) as contextual memory units. They differ because, following Sukhbaatar et al. (2019), they use a fixed array of key/values pairs that are treated as parameters and are determined at training time by gradient descent. Since these stored key/value pairs do not change at inference time, there is no need for a value extraction function $\psi(x_{T+1}, x_T, \dots)$

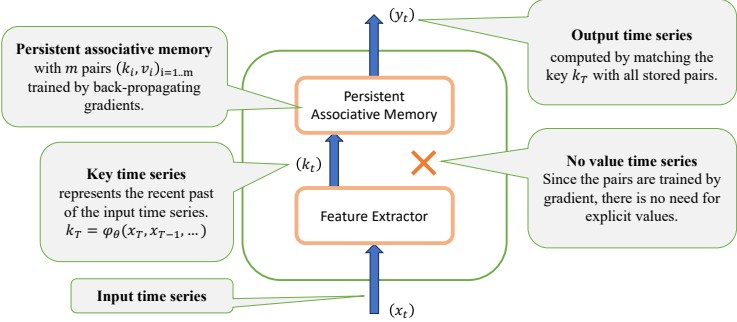

Figure 12: Persistent memory unit. The persistent associative memory contains a fixed number of key-value pairs $(k_i, v_i)_{i=1\dots m}$ whose values are determined by back-propagating gradients at training time. Since the memory contents do not change at inference time, there is no need for explicit values.

## D.2  TRAINING AND VALIDATION

Figure 13 plots the training and validation curves for both Transformer and Memory Mosaic in a manner similar to Figure 7 but showing additional block depths.

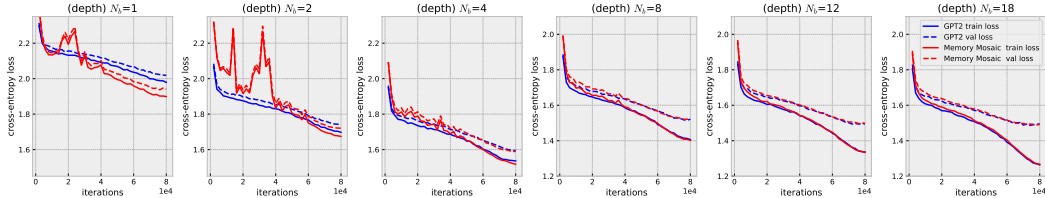

Figure 13: Additional training and validation curves for the transformer and Memory Mosaic architectures trained on BABISTORIES for more model depths than Figure 7.

Several comments can be made:

- The Memory Mosaic has a small advantage for very small depths ($N_b = 1$ and $N_b = 4$) but this advantage does not persist when the number of blocks increases. We believe this is due to the fact that a single layer Memory Mosaic can implement an induction head whereas a Transformer needs two layers. This amounts to saying that a $n$ block deep Mosaic has the same number of parameters than a $n$ block deep Transformer, its performance is closer to that of a $n + 1$ block Transformer. This is not much of an advantage when $n$ gets large.

- The Memory Mosaic training uses the hyper-parameters that worked best for the Transformer and operates on the same mini-batches of examples in the same order. However, for small block depths, the Memory Mosaic training curve shows initial instability, suggesting that it might benefit from a smaller stepsize.

- The similarity of the Transformer and Memory Mosaic curves is especially striking when one recalls that the Memory Mosaic does not use position encoding. In fact Memory Mosaic have two mechanisms for dealing with positions. The first one is the fact that the values

$v_T$ peek one time position ahead. The second one is the leaky integration in (6). These two mechanisms are useful to implement bigram or n-gram induction heads in a single layer, but they do not allow a head to selectively address a token by position (we use a single scalar leaky average coefficient per head). This suggests that position encoding in Transformers is mostly useful to implement an initial induction head in the first two blocks.

### D.3 QUALITATIVE EVALUATION

Table 5 provides a variant of Table 4 in Section 7, with $N_b = 1$.

### D.4 DIFFERENCES IN ATTENTION AND THE LEAKY AVERAGE COEFFICIENT $\lambda_\varphi$

Because Memory Mosaics lack position encoding and do not distinguish keys and queries, we investigate how their attention patterns differ from those of transformers. Figure 14 shows attention scores for each head of either a one-block deep transformer using absolute position encoding (left plot) or a one-block deep Memory Mosaic (right plot). The scores are averaged on 5000 BABIS-TORIES sequences and show how the last position attends to earlier positions in the 512 token long context window. The transformer attention patterns are noisy, with a strong "attention sink" at position 0 (Xiao et al., 2023). In contrast, the Memory Mosaic attention pattern is mostly flat, save for higher scores for the most recent tokens.[13]

Figure 15 show the attention patterns for contexts extended to 1536 tokens, using models trained on 512 token long sequences. Because the absolute position encoding scheme cannot be extended to longer contexts, we provides a comparison with transformers using RoPE (Su et al., 2024) and ALIBI (Press et al., 2022). The RoPE attention patterns do not extend nicely beyond the training context length. The ALIBI attention patterns show the vanishing contribution of distant tokens. In contrast the Memory Mosaic attention patterns remain mostly flat.

Figure 16 shows the relationship between attention map and leaky average coefficient $\lambda_\varphi$.

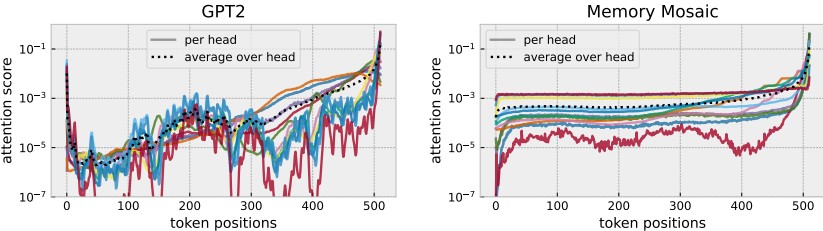

Figure 14: Average attention scores of the last token attending previous tokens (evaluated on an in-distribution validation dataset). Each solid line indicates one head in either the transformer attention block or the Memory Mosaic contextual memory block. The dotted line averages the attention of all heads. All models are trained with context length 512.

### D.5 IN-CONTEXT LANGUAGE LEARNING EVALUATION

Table 3 provides the IID test performance of various architectures trained on REGBENCH (Akyürek et al., 2024) with 100 training environments. We keep the training process, including hyperparameter searching space, to be the same as the one in Figure 9. But sample validation and test sets from the same 100 probabilistic finite automatons (training environments) as the training set. This table, together with Figure 9, show that *baseline methods learned the training environments (good IID) but not the meta-learning ability (poor OOD)*.

---

[13]This effect is connected to the leaky average coefficient $\lambda_\varphi$, as shown in Figure 16.

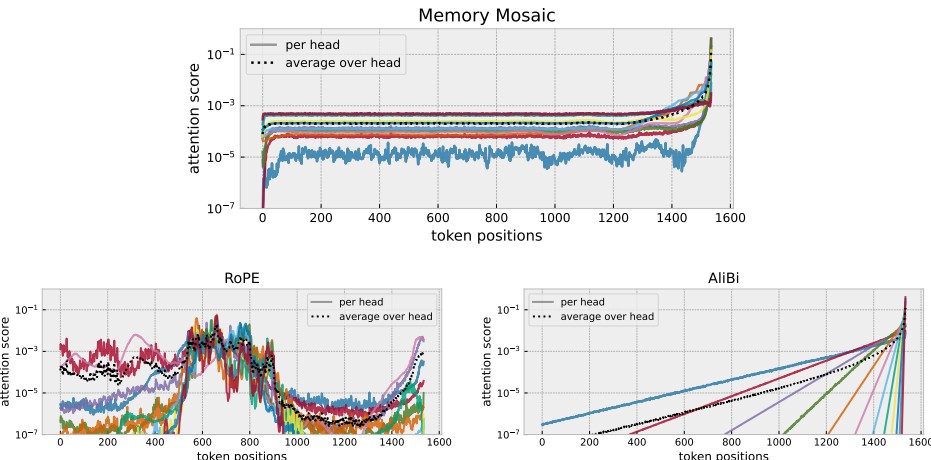

Figure 15: Average attention scores on an extended context window ($3 \times 512$ tokens). Models are still training with a 512 token long context window. Because the GPT2 absolute position encoding does not extend, we compare with RoPE (Su et al., 2024) and ALIBI (Press et al., 2022).

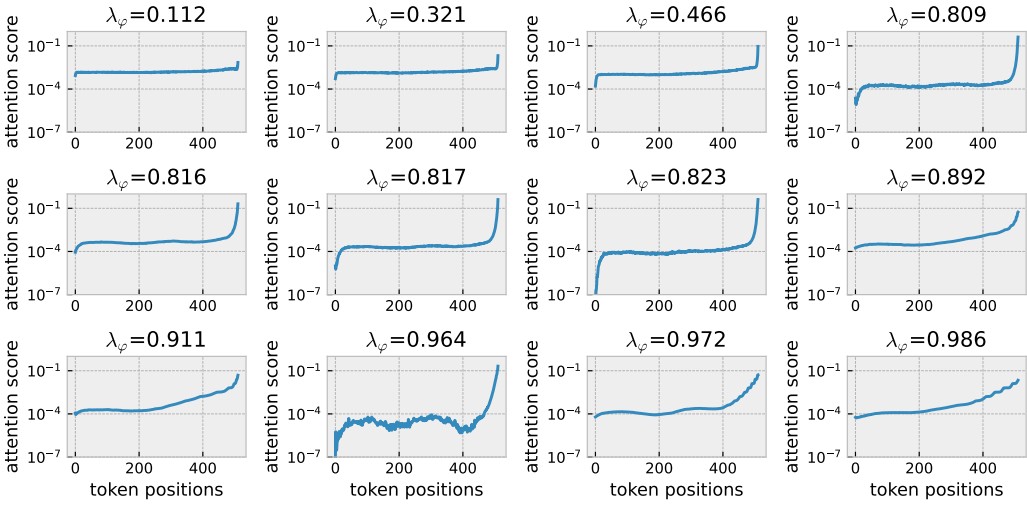

Figure 16: Attention map and leaky average coefficient $\lambda_\varphi$. As $\lambda_\varphi$ increases, $k_t$ in Eq 6 effectively takes a longer history into the account, and thus the pick at the end of attention map becomes wider.

Table 3: In-distribution (IID) performance of various architectures trained on REGBENCH (Akyürek et al., 2024) with only 100 training environments. Both training, validation, and test set (100 samples) are sampled from the same 100 random probabilistic finite automatons (PFA). Compared with the poor OOD accuracy ($\sim$0.45) / TVD ($\sim$0.75) of baseline methods in Figure 9, All baseline methods perform well in the IID test set (even with only 100 training environments).

|  | Memory Mosaic | tf | Mamba | S4 | RWKV | linear tf | H3 | GLA | Hyena | LSTM | RetNet |
|---|---|---|---|---|---|---|---|---|---|---|---|
| Accuracy (↑) | **0.959** | 0.856 | 0.929 | 0.846 | **0.967** | 0.816 | 0.794 | 0.870 | **0.953** | 0.849 | 0.876 |
| TVD (↓) | 0.417 | 0.308 | **0.268** | 0.350 | **0.183** | 0.348 | 0.425 | 0.284 | **0.244** | 0.343 | 0.296 |

# E  Computing Resources

Models were trained on 64 NVidia V100 GPUs over 80k epochs. From conception to finalization of this paper we trained about 200 models. To create the BabiStories dataset via Mistral, we ran with 128 NVidia V100 GPUs for 3 days. The supporting machines contain Intel(R) Xeon(R) Gold 6230 CPUs. The 3 moons result took negligible resources and were trained on Apple M1 laptops.

Table 4: Continuations generated for the 24 prompts designed by Eldan & Li (2023) to investigate the factual, reasoning, and consistency capabilities of language models trained on BABISTORIES. Both the transformer and the Memory Mosaic were $N_b = 18$ blocks deep. Colors green and red respectively indicate correct and incorrect answers. The text was generated with a beam search of width 2 and manually truncated to the shortest length that answers or fails to answer the question.

| Prompt | Transformer ($N_b = 18$) | Memory Mosaic ($N_b = 18$) |
|---|---|---|
| Alice was so tired when she got back home so she went | to sleep. | straight to bed. |
| Jack and Lily saw a rainbow after a rainy day. They were amazed by the colors. Jack said, "Look, Lily. A rainbow has | many colors!" | many colors!" |
| Jack and Lily liked to watch the moon at night. They noticed that the moon changed its shape every night. Sometimes the moon was big and round, and sometimes it was | small and thin. One night, Jack said, "Lily, look! The moon is changing." | small and thin. One night, Jack had an idea. He said, "Let's make a picture of the moon!" |
| Jack wanted to read a book, so he went to | the library. | the library. |
| "Can cows fly?", Alice asked her mother | Her mother laughed and said, "No, Alice. Only birds and planes can fly." | "No, dear. Only birds can fly," her mother replied with a smile. |
| "What do birds like to eat?", Tom asked his mother. | "Birds like to eat worms and bugs," his mother replied. | "Birds like to eat worms and bugs, Tom," his mother replied. |
| "What language do **they** speak in France?", Tom asked his mother | His mother smiled and said, "**We** all speak different languages, Tom. It's what makes us unique and special." | His mother smiled and said, "They speak in different languages, Tom. But we can learn some of them." |
| If I throw a ball up in the air, eventually it will | come down and I can catch it. | come down. |
| It was winter and cold outside so his mother told him, "You should | wear your warm coat, dear." | wear your warm coat and hat." |
| Lily likes cats and dogs. She asked her mom for a dog and her mom said no, so instead she asked | for **a toy**. | her dad. |
| Jack told Mary, "If you give me your banana, I'll give you my apple". Mary gave Jack her banana so | he could eat it. | that he could eat it. |
| On **weekends** Jack went to visit his **grandmother** whereas on **weekdays** he would go to **school**. Last **weekend**, when Jack was on his way to | **school**, | **school**, |
| Lily and Ben were having an argument. Ben said that cake is much better than ice cream and Lily said that | ice cream is the best. | she likes ice cream |
| Lily and Ben are having an argument. They are trying to decide between the park and the swimming pool. Ben says, "I want to go to the park". Lily says | , "No, | "No, |
| Jack's **mother was not home**, and his **father was at home**. When Jack came home, he said hello to | his **mother**. | his **mother** |
| Lily doesn't like swimming. When her father wants to take her to the swimming pool, she says | , "No, daddy. I don't want to go." | , "No, thank you." |
| Both Ben and Lily wanted cake. Father said that there was only one piece of cake left. They | both wanted to eat it, but there was only one. | all wanted to eat it, so they began to fight. |
| Ben went to visit Lily in her house, but she was not at home. Ben knocked on the door, | and Lily's mom answered. | and Lily's mom opened it. |
| "Hi Jane, have you seen Alice? I can't find her anywhere", said Jack. | Jane shook her head. "No, I haven't. But I will help you look." | Jane shook her head. "No, I haven't seen Alice. But I can help you look!" |
| Max had two dogs. One was **white** and the other was **black**. Max walked up the street and saw a kid with a dog. He told the kid, "I see you have a brown dog. I also have | a white dog." | a **brown** dog." |
| Anne had a piece of candy in her left pocket and a piece of chocolate in her right pocket. Anne's mom asked her, "Anne, what is that you have in your left pocket?" | Anne smiled and said, "It's a candy | Anne opened her hand to show her mom the candy |
| Alice had both an **apple and a carrot** in her bag. She took the **apple** out of the bag and gave it to Jack. She reached into the bag again and took | out the carrot. | out **another apple**. |
| Alice and Jack walked up the street and met a girl in a red dress. The girl said to them, "Hi, I'm Jane. What are your names?" | Alice replied, "I'm Alice. This is Jack. We came to see you." | Alice smiled and said, "I'm Alice. This is Jack. We are going to the park to play." |
| Diva was hungry, and wanted to bake a cake, but she didn't have any sugar at home, so she decided to go ask around. She started walking and met a squirrel. She asked the squirrel, "Would you happen | to know where I can find some sugar?" | to see any sugar around here?" |

Table 5: Continuations generated for the 24 prompts designed by Eldan & Li (2023) to investigate the factual, reasoning, and consistency capabilities of language models trained on BABISTORIES. Both the transformer and the Memory Mosaic were $N_b = 1$ blocks deep.

| Prompt | Transformer | Memory Mosaic |
|---|---|---|
| Alice was so tired when she got back home so she went | to sleep | to bed |
| Jack and Lily saw a rainbow after a rainy day. They were amazed by the colors. Jack said, "Look, Lily. A rainbow has | come!" | come to us!" |
| Jack and Lily liked to watch the moon at night. They noticed that the moon changed its shape every night. Sometimes the moon was big and round, and sometimes it was | just a big, round moon. | too big. |
| Jack wanted to read a book, so he went to | find his friend, a wise old owl named Hoot. | find his best friend, Timmy. |
| "Can cows fly?", Alice asked her mother | Her mother smiled and said, "Of course, dear. | Her mother smiled and said, "Sure, honey. |
| "What do birds like to eat?", Tom asked his mother. | "I don't know, but it looks yummy!" Tom replied. "Let's find out!" | His mother smiled and said, "That's a good idea, Tom. |
| "What language do they speak in France?", Tom asked his mother | France smiled and said, "It's a language, Tom. It helps us talk to each other." | "Mom, what is that?" France asked. |
| If I throw a ball up in the air, eventually it will | go far away. | come back. |
| It was winter and cold outside so his mother told him, "You should | go outside and play in the snow." | stay in your warm, warm home." |
| Lily likes cats and dogs. She asked her mom for a dog and her mom said no, so instead she asked | , "Can we keep it, mom?" Her mom said, "Yes, but we need to keep it safe." | her mom if they could keep the dog. Her mom said yes, and they named the dog Sparky. |
| Jack told Mary, "If you give me your banana, I'll give you my apple". Mary gave Jack her banana so | he could have it. | she could eat it. |
| On weekends Jack went to visit his grandmother whereas on weekdays he would go to school. Last weekend, when Jack was on his way to | school, | school, |
| Lily and Ben were having an argument. Ben said that cake is much better than ice cream and Lily said that | they could both have fun. | it was the best cake they ever had. |
| Lily and Ben are having an argument. They are trying to decide between the park and the swimming pool. Ben says, "I want to go to the park". Lily says | , "No, | "No, |
| Jack's mother was not home, and his father was at home. When Jack came home, he said hello to | his mother, | Jack. |
| Lily doesn't like swimming. When her father wants to take her to the swimming pool, she says | "Lily, you can do it! | , "Mommy, I want to swim too!" |
| Both Ben and Lily wanted cake. Father said that there was only one piece of cake left. They | all sat down to enjoy the yummy treat. | both felt sad. |
| Ben went to visit Lily in her house, but she was not at home. Ben knocked on the door, | and the door opened. A kind lady came out | and when **Ben** opened the door, |
| "Hi Jane, have you seen Alice? I can't find her anywhere", said Jack. | Alice smiled and said, "Sure, I will help you find your way home." | "I don't know, Jack. |
| Max had two dogs. One was white and the other was black. Max walked up the street and saw a kid with a dog. He told the kid, "I see you have a brown dog. I also have | a black dog." | a brown dog." |
| Anne had a piece of candy in her left pocket and a piece of chocolate in her right pocket. Anne's mom asked her, "Anne, what is that you have in your left pocket?" | Anne smiled and said, "Yes, mommy. I found it in the park." | Anne smiled and said, "I found it on the ground. It's mine!" |
| Alice had both an apple and a carrot in her bag. She took the apple out of the bag and gave it to Jack. She reached into the bag again and took | out the apple. | out the apple. |
| Alice and Jack walked up the street and met a girl in a red dress. The girl said to them, "Hi, I'm Jane. What are your names?" | Jane smiled and said, "I'm **Timmy**, | Jane replied, "I'm Jane. |
| Diva was hungry, and wanted to bake a cake, but she didn't have any sugar at home, so she decided to go ask around. She started walking and met a squirrel. She asked the squirrel, "Would you happen | to my house, little one?" | to my cake?" |

