# OpenReview forum: "Memory Mosaics"
_ICLR.cc/2025/Conference — ICLR 2025 Poster_

### Official Review · Reviewer_9KrW · 2024-10-17

**Soundness:** 2
**Presentation:** 2
**Contribution:** 2
**Rating:** 6
**Confidence:** 3

**Summary:**

This paper proposes Memory Mosaics, a neural network architecture composed of associative memory units for language modeling tasks. The authors present the concept of "predictive disentanglement" to explain how the model decomposes prediction tasks. Experiments show comparable or better performance to transformers on language modeling benchmarks, with claimed advantages in interpretability and in-context learning.

**Strengths:**

-  The paper addresses an interesting problem in developing more interpretable language models.
- The work is well-organized and clearly presented.

**Weaknesses:**

- The core concept presented in Figure 3 bears similarities to Linear Transformers [1], particularly RetNet [2] (see RetNet Eq. 6). However, the paper fails to adequately discuss or cite these highly relevant works. Linear Transformers are only briefly mentioned in Figure 9, without proper citation or in-depth comparison. This oversight is particularly concerning as it appears the authors are aware of the similarities, given the inclusion of Linear Transformer, RetNet, and GLA [3] in Table 3 and Figure 9. The lack of a detailed explanation comparing their approach to the evolution of Linear Transformer methods (such as GLA/RetNet) raises questions about the paper's originality and positioning within the field.

- The authors introduce persistent memory units (P-mem) but do not clearly differentiate them from standard MLP layers. They state that P-mem can be seen as MLP layers (lines 328-330), but fail to elucidate the advantages or unique properties of P-mem that justify its introduction as a new concept. This raises concerns about the necessity and value of creating this new terminology.

- While the paper uses RegBench to assess in-context learning, this benchmark alone may not provide a comprehensive evaluation of the model's memory capabilities. The performance of state-space models like Mamba can be significantly impacted by the inclusion of short convolutions, as noted in DeltaNet [4] (See Figure 2). A more thorough evaluation, possibly including benchmarks like MQAR [5], would provide a more robust assessment of the model's capabilities relative to existing approaches. I understand that there may be not enough time for running such experiments in the rebuttal stage, so the author can just try to rerun run the experiment with/without the short convolution in RegBench.

- The paper focuses primarily on language modeling tasks but lacks evaluation on standard common reasoning benchmarks that are crucial for assessing language models' capabilities. Tasks such as ARC Challenge, ARC Easy, PIQA, and Hellaswag are widely used to evaluate smaller-scale language models and provide valuable insights into a model's reasoning abilities. The absence of these evaluations limits our understanding of the model's true capabilities and makes it difficult to compare with other approaches in a comprehensive manner. Including these benchmarks would provide a more holistic view of the model's performance across various aspects of language understanding and reasoning.

- The paper appears to present an incremental advance on linear transformers, repackaged with new terminology and concepts. While this approach creates an impression of novelty, its actual contribution to advancing the field seems limited. The work seems to complicate a relatively straightforward evolution of linear transformer models with complex conceptual packaging, which may hinder rather than help the development of the research community and the broader field.

If the authors can appropriately address my concerns or correct me if I am wrong, I would be willing to increase my score.

[1] Katharopoulos, Angelos, et al. "Transformers are rnns: Fast autoregressive transformers with linear attention." International conference on machine learning. ICML, 2020.

[2]Sun, Yutao, et al. "Retentive network: A successor to transformer for large language models." arXiv preprint arXiv:2307.08621 (2023).

[3]Yang, Songlin, et al. "Gated Linear Attention Transformers with Hardware-Efficient Training." ICML, 2024.

[4]Yang, Songlin, et al. "Parallelizing Linear Transformers with the Delta Rule over Sequence Length." NeurIPS, 2024.

[5]Arora, Simran, et al. "Simple linear attention language models balance the recall-throughput tradeoff." ICML, 2024.

**Questions:**

See my weakness part.

---

> ### Author Response · Authors · 2024-11-15
>
> Please read our general responses, especially about the (non)-relation to Linear Transformers.
>
> 1) *The core concept presented in Figure 3 bears similarities to Linear Transformers [1], particularly RetNet [2] (see RetNet Eq. 6). However, the paper fails to adequately discuss or cite these highly relevant works*
>
> As explained in the general response, our work is entirely orthogonal to work on Linear Transformer. In particular, equation (7) makes clear that the computational cost is quadratic, like that of an ordinary transformer. The only connection to Linear Transformer are the engineering opportunities mentioned as future work in the last paragraph of the conclusion.
>
> 2) *The paper appears to present an incremental advance on linear transformers, repackaged with new terminology and concepts. While this approach creates an impression of novelty, its actual contribution to advancing the field seems limited. The work seems to complicate a relatively straightforward evolution of linear transformer models with complex conceptual packaging, which may hinder rather than help the development of the research community and the broader field.*
>
> We can affirm that this paper does not constitute an incremental advance on linear transformers, because it is not about linear transformers at all.
>
> 3) *The authors introduce persistent memory units (P-mem) but do not clearly differentiate them from standard MLP layers. They state that P-mem can be seen as MLP layers (lines 328-330), but fail to elucidate the advantages or unique properties of P-mem that justify its introduction as a new concept. This raises concerns about the necessity and value of creating this new terminology.*
>
> Please check our general response 3.
>
> 4) *While the paper uses RegBench to assess in-context learning, this benchmark alone may not provide a comprehensive evaluation of the model's memory capabilities. The performance of state-space models like Mamba can be significantly impacted by the inclusion of short convolutions, as noted in DeltaNet [4] (See Figure 2). A more thorough evaluation...*
>
> Apart from the Memory Mosaic results, all results in Figure 9 were obtained by Akyurek et al. in the RegBench paper. Their code uses the publicly available implementations in a very well organized way. Since our work is not related to Linear Transformer and variants, we do not see a reason to relitigate the Regbench results. Note however that there is a regime in which the Mamba and H3 performances exceeds that of transformers, but, even in this regime, both are much worse than Memory Mosaics (and that is our point).
>
> 5) *The paper focuses primarily on language modeling tasks but lacks evaluation on standard common reasoning benchmarks that are crucial for assessing language models' capabilities. Tasks such as ARC Challenge, ARC Easy, PIQA, and Hellaswag are widely used to evaluate smaller-scale language models and provide valuable insights into a model's reasoning abilities. The absence of these evaluations limits our understanding of the model's true capabilities and makes it difficult to compare with other approaches in a comprehensive manner. Including these benchmarks would provide a more holistic view of the model's performance across various aspects of language understanding and reasoning.*
>
> Glad you asked.  At the relatively small GPT-2 scale, we find that these benchmarks tell very little about the actual model performance at scale. See for instance the recent Apple paper (https://arxiv.org/pdf/2410.05229) revealing the brittleness of such "reasoning" benchmarks.
>
> We like the TinyStories approach (https://arxiv.org/abs/2305.07759) because it provides the means to probe large language modeling concerns in small models. We find that a deep dive into the TinyStories/BabyStories performance is a much better predictor of the general language modeling performance of a similar model on a much larger scale. Our ongoing scaling work certainly agrees with this claim made in the TinyStories paper.

---

> ### Comment · Reviewer_9KrW · 2024-11-21
> **Thank you for your rebuttal.**
>
> The reviewer appreciate authors' thorough and detailed responses to the concerns raised in the review. After carefully going through all your comments and clarifications, I find your rebuttals regarding the distinction from Linear Transformers and the other technical aspects to be well-reasoned and convincing.
>
> As my primary expertise lies in model pre-training rather than the core focus of this paper, I acknowledge that I may have misinterpreted some of the technical aspects initially. However, I still have one remaining concern regarding the evaluation benchmarks. While the provided link discusses the limitations of mathematical reasoning benchmarks, my original comment referred to commonsense reasoning benchmarks (ARC, PIQA, HellaSWAG). These benchmarks have been shown to have a positive correlation with language modeling loss scaling laws [1], suggesting their validity even for smaller-scale models. I believe evaluating on these benchmarks would still provide valuable insights into your model's capabilities.
>
> Given your thorough clarifications on the technical aspects and architectural decisions, I am revising my score upward.
>
> [1]: Du, Zhengxiao, et al. "Understanding emergent abilities of language models from the loss perspective." arXiv preprint arXiv:2403.15796 (2024).

---

> ### Author Response · Authors · 2024-11-21
>
> We thank you for accepting to re-examine your initial assessment and we commend you for your intellectual integrity doing so.
>
> Regarding the benchmarks, please note that we do not claim that memory mosaics should replace transformers in LLMs at this time, we merely claim that they give enlightening insights in the how this kind models work, and that they offer opportunities that might prove useful.  We believe that our limited benchmarks are adequate for this claim:
>
> * First let's look at the minimum kind of evidence needed to claim that memory mosaics should actually replace decoding transformers for big natural language applications. Just to be competitive for the kind of standard benchmarks you are suggesting, we would need scale our models to a couple billion parameters trained on a generic language corpus. We would have to get serious about hyper parameter search, establish corresponding scaling laws, etc.  We would also need to demonstrate substantial new capabilities, such as for instance show that our models have uncharted in-context learning ability (in the spirit of Fig 9). But that might not be suitable for all applications since superior combinatorial learning abilities could also make the models more willing to hallucinate, and we would have to sort that out. This an entirely new research effort, certainly worthwhile, but only possible as a second step. Setting such a bar for any research on transformer ideas would prevent a lot of research from happening.
>
> * Since our paper focuses on the insights gained by our transformer-like setup, we only need to show when our model behaves like transformers (e.g. fig 7, tables 4 and 5) and when it differs in interesting ways (e.g. Fig 8, Fig 9, Fig 15). This is more a qualitative comparison than a quantitative one.  These comparisons are not precise enough to be changed into scaling laws, but they are still informative enough for our claim.
>
> As an aside, we carefully read the reference you suggest and in particular the way experiment distinguish "easy" benchmarks like HellaSwag from "hard" benchmark like GSM8k. We also notice that the "easy" benchmarks often involve selecting one of n preselected continuations which is far easier (and also far easier to measure) than letting the model produce free generations. In contrast, note how Tables 4 and 5 show free generations using quite small models (<160M). The quality of these free generations cannot be matched by much larger models (1.5GB params as discussed in the TinyStories paper) trained on a general language corpus, even though these larger models can perform well on HellaSwag-style benchmarks. All this to say that the tiny stories setup gives very interesting qualitative information, albeit not the kind you could use to obtain scaling laws for instance. Yet, for the objectives of our paper, this works.

---

> > ### Comment · Reviewer_9KrW · 2024-12-02
> >
> > After carefully reviewing the authors' detailed rebuttal, I am convinced of their strong technical background and the solid theoretical foundation of their work. The comprehensive responses demonstrate the rigor and depth of their research. Given these considerations, I would like to revise my score to 6, as I believe this work makes a valuable contribution to the field. I particularly appreciate the authors' thorough and thoughtful responses in the rebuttal, which clearly demonstrate their dedication to this research.

---

### Official Review · Reviewer_k2DW · 2024-10-22

**Soundness:** 3
**Presentation:** 3
**Contribution:** 3
**Rating:** 8
**Confidence:** 4

**Summary:**

This paper introduces a new neural network architecture, the Memory Mosaics, which employ associative memories to complete prediction based tasks. The Memory Mosaic architecture consists of a number of "elementary memory units", which in turn consist of a feature extractor (outputting a time series of "key" vectors and a time series of "value" vectors), which feeds an associative memory. Prediction in the Memory Mosaic has the associative memories update based on new information from the task, rather than having weights frozen from some previous training, such as in traditional deep feed-forward neural networks. The authors link the Memory Mosaic architecture to the transformer architecture (in one specific example in Section 6, GPT-2) and the attention mechanism. This link is derived from Gaussian kernel smoothing and enforcing key vectors to have equal magnitude. To differentiate this work from other works linking associative memories and attention (e.g. Ramsauer et al. 2020) this paper focuses on the ability of the Memory Mosaic to "peek" one step ahead in prediction tasks. The paper asserts that the learning process of Memory Mosaics can be framed as a meta-learning process. The meta-learning process selects for associative memories that are faster at learning new tasks, and hence can produce useful value estimates in fewer time steps. The authors continue to discuss "predictive disentanglement"; a framework in which discrete elementary memory units may learn discrete aspects of the prediction task, allowing for interpretable networks with modules of definitive responsibilities (at least, for tasks that can be easily disentangled). The authors demonstrate predictive disentanglement with a toy predictive task based on modelling the orbit of three moons (represented as a time series of positions). In the presented example, the Memory Mosaic is capable of separating the independent orbits of each moon, allowing the network to make more accuracy predictions earlier, i.e. before all moons have finished an orbit. The next example tests the Memory Mosaic in language based tasks. The authors present an altered memory unit that has architectural similarities to the GPT-2 transformer. The authors create a new dataset based on TinyStories (Eldan & Li 2023) titled BabiStories. BabiStories is generated from the Mixtral-8x7B language model and is otherwise similar to the TinyStories dataset, except with an increased diversity of stories and requiring first names and specific opening words for each story. Memory Mosaics demonstrate improved losses compared to GPT-2 for small networks, but has strikingly similar performances for deeper networks. Out-of-distribution language-based experiments were conducted on the Simple English Wikipedia dataset, in which Memory Mosaics appear to perform much better than GPT-2. Further experiments are performed using RegBench, in which Memory Mosaics appear to outperform several state-of-the-art architectures in both accuracy and total variation distance.

**Strengths:**

The paper presents an interesting, seemingly novel architecture for a prediction model based on associative memories. The architecture seems well justified from first principles and links nicely with existing works on the attention mechanism. This work is of good quality and presents its ideas reasonably well, including rigorous formalizations and numerous figures explaining the proposed architectures. The paper benchmarks the Memory Mosaic against a toy dataset to demonstrate properties of the network, as well as several language-based datasets which allow for comparisons with transform architecture models. These benchmarks add to the quality of the paper, and the newly introduced BabiStories benchmark is original.

**Weaknesses:**

To my reading, the architecture is not differentiated strongly enough from existing, similar work on associative memories and the attention mechanism. The key property of the architecture, "peeking" in the value predictions, is not explained clearly enough for a reader to understand the significance. The meta-learning interpretation in Section 3 is somewhat confusing in how this relates to the broader field of meta-learning. The toy dataset, while very useful in understanding "predictive disentanglement", appears to be crafted specifically to show this property and it is not clarified if such a property is present in other, non-toy datasets. The paper introduces a new language-based dataset, BabiStories, which is similar to the existing TinyStories, but it is not immediately apparent (from the main text or the appendices) why a new dataset is required. Further, without benchmarking of the introduced BabiStories dataset with several models it is not clear if the dataset is useful in assessing the performance of the Memory Mosaic.

**Questions:**

Equation 2 offers a nice, specific implementation of a conditional expectation set forth in Equation 1. The discussion here flows very smoothly and is easy to understand. However, the exact form of Equation 2 --- the Gaussian kernel smoothing --- bears striking similarity to the softmax distribution (and in turn the attention mechanism, as the authors discuss later) as well as the Boltzmann distribution. From statistical physics it is well known that the normalizing factor (Z in Equation 2) is nearly always intractable for large enough systems. In the context of Memory Mosaics, it is not immediately apparent what this would correspond to, but it would certainly be of interest to discuss where this implementation would stumble based on this factor. The authors do sidestep the intractability issue in Equation 3 by asserting that the magnitude of the key vectors (rather, the squared norm) are equal, which nicely reduces Equation 2. Again, it is not immediately clear if that is always going to be the case for Memory Mosaics, if it will always be possible to enforce this constraint in applications, or if this only occurs in the datasets presented in the paper. In any case, a brief discussion on the possibility of non-uniform squared norm key vectors may be of interest to readers and help ground the Memory Mosaic in the landscape of prediction models. The discussion on why this paper limits itself to only Gaussian kernel smoothing derived associative memories is excellent; clear and concise. Emulating this level of discussion for these other points may prove helpful. In short: is the calculation of the normalizing factor Z as intractable as it appears? What factor of the Memory Mosaic / dataset makes the normalization factor impossible to calculate? Does the uniform-squared-norm trick in Equation 3 work for all datasets, or is it a rare occurrence that makes the network otherwise untenable?

The authors discuss "peeking" in the Memory Mosaic, which allows the memory units to look one time step ahead in the "key" series to predict the next item in the "value" series. After several read-throughs, it is still difficult to understand what this means for the Memory Mosaic. In a more traditional prediction model looking a timestep ahead would be considered equivalent to a bug. In the Memory Mosaic this is seemingly not the case, but the reasoning is still hazy to me. Am I correct in my understanding that "peeking" is allowable in this model, and if so, why is this distinct from other associative memories? Is it possible to clear up the discussion on this point, particularly the subsection in Section 2, to explain the nuances of "peeking" more? Considering the importance of the "peeking" mechanism in this work, it is paramount it is easy to digest.

The Memory Mosaic architecture has striking similarities to associative memories linked to attention mechanisms. In particular, Ramsauer et al.'s influential "Hopfield is All You Need" has almost the same derivations found in Section 2 of this paper. The authors discuss the Ramsauer paper at the end of Section 2, but the differentiation of the works does not appear substantial. In particular, the "peeking" of the Memory Mosaic is not well explained, which makes the reliance on it for the basis of novelty compared to Ramsauer et al. troublesome. With some further discussion of the "peeking" mechanism (as noted above), and potentially some more clarity on the section discussing Ramsauer et al., this paper would provide a much stronger foundation for Memory Mosaics in the context of existing literature.

The meta-learning process at the start of Section 3 is somewhat confusing when primed to think about both meta-learning as a field and traditional training of neural networks. If my understanding of the paper is correct, training a neural network (e.g. writing a training loop, feeding in training data, performing backpropagation, updating weights, and so on) is presented here as meta-learning, with the "regular" learning being the updates to the associative memories performed when prediction of a time series occurs. This is a fairly radical change in perspective, although an interesting interpretation, but again it seems that the paper could benefit from having this stated very directly as to guide the reader to this way of thinking about meta-learning in the Memory Mosaic. Is this interpretation of meta-learning in the Memory Mosaic correct? If so, the steam-roller metaphor (Figure 2) gives rise to another interesting question that is seemingly not addressed in the main body of the paper; does the Memory Mosaic always benefit from fast-learning memory units? Certainly I can see the benefit of having a predictor model that requires a smaller context window to start spitting out predictions, but is it possible that slow-learning memory units result in more accurate predictions (for any number of reasons). If that is the case, it would raise concerns about the gradient / meta-learning presented in the paper, and if that is not the case I have not been convinced by the text. Is it correct to say that faster-learning memory units in the context of Memory Mosaics are always better?

The toy example presented --- the three moons problem --- is excellent at showing the application of "predicative disentanglement". However, it is less clear after reading through the main text and the appendices if this specific kind of "easy" disentanglement is present in other, non-toy datasets. Mining this sort of information from, say, a language-based dataset would be nigh impossible, but some discussion about whether the property is present in other datasets (or even if this is likely to occur) would make for a nice conclusion to the Section. Moreover, the authors mention several times that the weight matrices in the three moons problem could be replaced with the identity to improve results. In Appendix A these matrices are seemingly the anti-diagonal identity. Is the correct interpretation here that the rows of these matrices could be permuted freely, and hence are equivalent to the identity? Otherwise, these do not seem to be identity matrices, which makes the previous discussion odd!

The main language-based task presented in this paper is based on BabiStories. The new dataset is seemingly introduced without explanation as to why it is needed. Due to the similarities to TinyStories, it seems that BabiStories must include some additional structure that makes it useful when working with Memory Mosaics in particular, but this is not stated clearly in the main text. Appendix B may mention this briefly, but it is not clear if the dataset is created for this reason or not. Is it possible to use TinyStories to train Memory Mosaics and use these results in place of BabiStories? If not, perhaps a description of why not could be included in the main text to explain to the reader why a new dataset is required.

Figure 7 shows the performance gap between the Memory Mosaic and GPT-2 models when trained on the BabiStories dataset. These are promising curves and show excellent results. However, this performance gap is only substantial for a model depth of 1. Even at a model depth of 8, the second sub-figure, the gap between GPT-2 and Memory Mosaics has seemingly disappeared. It is still interesting that GPT-2 and Memory Mosaics perform equally well for larger depths, does this hint that perhaps the equivalent performance is due to the dataset rather than the architecture i.e. both architectures can saturate to the best possible accuracy on BabiStories? Also, is it possible to focus on specifically the region where there is a difference in performances? The jump from a depth of 1 to a depth of 8 in this Figure seems to leave out the most interesting region in the context of the paper. Does the Memory Mosaic continue to outperform GPT-2 at a depth of 2 or 3, or do the two architectures converge immediately after a depth of 1? The hyperparameter transfer from GPT-2 to Memory Mosaics is also terrifically interesting --- considering how fickle these networks can be, and the equal performance show in Figure 7, does this hint at some deeper connection between Memory Mosaics and the GPT-2 architecture, perhaps even a full equivalence of the models? This last question is less a review of the current manuscript and more of a potential lead into future research, so no need to address this point in your comments!

---

> ### Author Response · Authors · 2024-11-15
>
> Many thanks for your very detailed review.
>
> 1) *From statistical physics it is well known that the normalizing factor (Z in Equation 2) is nearly always intractable for large enough systems.*
>
> The normalization factor in equation 2 is a sum of n terms where n is the number of key/values pair stored in the memory, which is equal to the number of past tokens visited in the input sequence, and always inferior to the maximal context length. This number of terms is perfectly tractable.  The normalization of the key vectors does not change it.
>
> 2) *The authors discuss "peeking" in the Memory Mosaic, [...] ahead would be considered equivalent to a bug.*
>
> This is indeed a key component of our architecture with nontrivial consequences. Please check our general response 2 for a rephrasing of this mechanism and its consequences. When presenting our work in front of an audience, we found it useful to spend more time on the explanation in lines 99-102 of the paper. Maybe these lines should be highlighted more prominently.
>
> 3) *If my understanding of the paper is correct, training a neural network [...] is presented here as meta-learning, with the "regular" learning being the updates to the associative memories performed when prediction of a time series occurs. This is a fairly radical change in perspective,*
>
> This is exactly what we mean (see lines 155-159) and this interpretation is critical for our paper.
>
> 4) *Does the Memory Mosaic always benefit from fast-learning memory units? [...] is it possible that slow-learning memory units result in more accurate predictions (for any number of reasons).*
>
> This is a good question. What we observe is that the (meta)-training process wants fast-learning memory units, and therefore constructs disentangled memory units. It might be that this is not the optimal way to model sequences that strictly follow the training set distribution. However, this property comes handy when the input sequence is out-of-distribution because it allows the network to quickly adapt to this new distribution. See Figures 8 and 9 for examples. This is key to in-context learning which itself plays an important role in the performance of transformer-like architectures.
>
> 5) *...  if this specific kind of "easy" disentanglement is present in other, non-toy datasets. Mining this sort of information from, say, a language-based dataset would be nigh impossible, but some discussion about whether the property is present in other datasets (or even if this is likely to occur) would make for a nice conclusion to the Section*
>
> We actually discuss this point lines 361-367 in the beginning of the language Section.  In addition, mining this sort of information from text might in fact be easier because, unlike moons, language was constructed to be easily learnable (an old argument in structural linguistics).
>
> 6) *Is the correct interpretation here that the rows of these matrices could be permuted freely, and hence are equivalent to the identity? [In the three moons network]*
>
> This is correct. The network architecture is symmetric with respect to an arbitrary permutation of the three heads.
>
> 7) *The main language-based task presented in this paper is based on BabiStories. [...] Due to the similarities to TinyStories, it seems that BabiStories must include some additional structure that makes it useful when working with Memory Mosaics.*
>
> We would have preferred to use the original TinyStories dataset. The unfortunate legal reason for constructing a new one is evoked line 376. To our knowledge, there is nothing in the BabiStories dataset that favors Memory Mosaics. Instead we went to some length to ensure that the BabiStories dataset behaves like the TInyStories dataset.
>
> 8) *The performance gap is only substantial for a model depth of 1. [...] It is still interesting that GPT-2 and Memory Mosaics perform equally well for larger depths, does this hint that perhaps the equivalent performance is due to the dataset rather than the architecture i.e. both architectures can saturate to the best possible accuracy on BabiStories?*
>
> The gap at model depth 1 is explained in footnote 6. What these plots show is that Memory Mosaics (without hyperparameter search) essentially match Transformers for in-distribution performance.  However, surprisingly strong differences appear with out-of-distribution inputs as in Figure 8 and 9.
>
> 9) The hyperparameter transfer from GPT-2 to Memory Mosaics is also terrifically interesting
>
> We do not believe that the GPT2 optimal hyperparameters are optimal for Memory Mosaics. We simply find it too easy to manipulate the performance by working on the hyperparameter search. Therefore we chose a protocol that completely avoids this risk.

---

> ### Comment · Reviewer_k2DW · 2024-11-18
> **Reply to Authors Comment**
>
> Thank you for your response! Your general responses above are well written and add to my understanding of your work. I acknowledge due to the length constraints placed by ICLR this information could not be placed into the paper, but it would be nice to have some of these explanations presented alongside the work you have done.
>
> The concept of peeking is starting to come in to focus for me, especially with your general response #2. As you say, this section may need to be rephrased to come across clearly to a new reader.
>
> I am still not sure if I agree with the terminology used in regards to "meta-learning" and "learning" in this paper. The convention in machine learning is of course to refer to the training of a model with data *before* inference time as "learning", although this work is somewhat different as this phase is "only" updating parameters and not memories. I appreciate there are two distinct phases of learning here, but I wonder if there is better terminology that could be applied to make it easier for a reader to understand what is happening in each phase.
>
> It would be unreasonable to request that you investigate the usefulness of "slow-learning" memories in the Memory Mosaic considering the quantity of research already present, but certainly this is something to look at in the future!
>
> I am not convinced that the same predictive disentanglement property discussed, and shown in the "three moons" toy problem, is present in language data. Lines 361 through 367 give some discussion on why the structure of some language tasks may be compatible with predictive disentanglement, but this does not constitute hard evidence to me. I could be lacking in my understanding of predictive disentanglement, but ideally it would be nice to see some kind of visualization or very concrete data on why language tasks have the same properties as the "three moons" problem.
>
> I see Line 376 is the motivation behind BabiStories. How unfortunate that legal issues handicap the use of the dataset! I stand by my point that introducing a completely new dataset is perhaps excessive alongside the remainder of the author's work --- it is difficult to gauge how useful this dataset is without benchmarking on many different models, which of course could be a paper in itself.
>
> I appreciate that the memory mosaic only requires one head to implement induction heads, whereas transformers require two. However, Figure 7 seems to imply (or at least suggest) that the Memory Mosaic outperforms transformer architectures for a depth up to 8. Is a depth of 8 when Memory Mosaics and transformers *start* to perform equally well, or does this occur at much smaller depths?
>
> As for the hyperparameter choices, I appreciate you are showing that the Memory Mosaic performs well even without tuning (you are giving transformers the upperhand and so on), but it is eyebrow-raising that using the same hyperparameters gives nearly exactly the same performance in Figure 7. This would suggest to me either the models have some underlying equivalence (you are observing the same behavior in two different expressions) or the models are both reaching some upper limit of the dataset. Moreover, I would still be interested to see hyperparameter tunings for the Memory Mosaic, to observe how much further they can be pushed.
>
> Thank you again for your reply.

---

> > ### Author Response · Authors · 2024-11-18
> >
> > Thanks for your updated comments.
> >
> > -----
> >
> > ### *On figure 7*
> >
> > We can provide an expanded version of figure 7 in the appendix. The results can be summarized as follows: although a n blocks deep mosaic has the same number of parameters as a n blocks deep transformer, it performs more like a n+1 block deep transformer. When n=1, this shows. When n grows, it quickly becomes moot.  The fact that the curves are so similar is related to using the same step sizes and the same training batches. If we change the step sizes or the batches, we can see more air between the curves although the final result is about the same.
> >
> > ------
> >
> > ### *On predictive disentanglement in natural language*
> >
> > There is another way to look at this besides the hints line 361-367.
> >
> > Suppose you're incrementally writing a little story using your favorite chatbot. You enter a first story, then repeatedly introduce new ideas and ask the model to reprint the whole story.  By doing so, you can build a story that is arbitrarily far from the training data, in the distant tail of the distribution, very far from any training example. Yet the system keeps producing syntactically correct language and reasonably coherent storylines. This means that the chatbot has discovered some of the compositional properties of language and is able to recombine different pieces of information coming from either the context or the training data. Understanding why and how this happens is a trillion dollar question. Some people say that it comes from scale only and that there is nothing to understand, but we don't have to believe this.
> >
> > Now look at figure 5, the three moons with three heads. Between step 150 and 250, the configuration of the three moons is very different from anything seen in the context (which contains only a small subset of the possible configurations) or in the training set (different moon periods). Yet the system has learned to disentangle the moons so that it can predict the future configurations by combining predictions for each of the moons. This conceptually similar to what you can see when you take a chatbot into the distant tail of its training distribution. This is far from a trivial piece of understanding, but a promising crack in the shell of the trillion dollar coconut.
> >
> > Shouldn't there be room in a conference for something like this?
> >
> > Anyway, if you think so, we'd be grateful if you considered upping your score...

---

> > > ### Comment · Reviewer_k2DW · 2024-11-19
> > > **Response to Authors**
> > >
> > > Thank you again for your reply.
> > >
> > > On Figure 7, I believe the current presentation of the figure (and surrounding discussion) does not capture what you have described in your previous comment. In my reading I took Figure 7 to mean that Memory Mosaics outperformed transformers at small depths, but the two architectures quickly converged to the same performance. Perhaps some form of your above comment could be useful in the paper? Explaining clearly that an n block memory mosaic performs like an n+1 block transformer could help a reader understand the takeaway from Figure 7. As for why the curves are so similar, I am not sure if I understand your explanation. Figure 7 shows cross entropy loss against iterations (which I am taking to mean epochs over the entire dataset). If that is the case, I would not expect the choice of step size or training batches to impact the results to a significant extent. It still seems to me like the two architectures are reaching some fundamental limit of the dataset based on how they respond to increased training epochs.
> > >
> > > On predictive disentanglement, your explanation above is nice, but I still am not convinced that the same disentanglement of the moons (a very pure, mathematical decomposition) can be applied to natural language. This could be my own unfamiliarity with natural language tasks, but I believe a reader would benefit from more explanation here. Is there any literature you could cite to show that natural language is easily disentangled? Is this a well known fact of natural language tasks? This is likely the most important change I would ask for in this manuscript, although it also requires the most work.
> > >
> > > I believe this work is interesting and should be discussed at a conference, but requires the above fixes to make it clear that the results are as significant as they seem.

---

> ### Author Response · Authors · 2024-11-20
>
> Please check our revised paper (which is now accessible through the usual pdf link in open-review).
>
> * We have added a section in the appendix that augments the material in Figure 7 and discusses it in detail.  Note that we do not claim that memory mosaic outperform transformer in these in-distribution evaluations. We only claim that it closely matches them despite the absence of position encoding, for instance. Figures 8 and 9 then provide out-of-distribution situations in which memory mosaics perform better.
>
> * In order to remind the reader that the compositional nature of language is essentially a mathematical property, we have added a reference to the 1968 book of Zellig Harris "Mathematical Structures of Language" which non only formalizes how sentences can be built through well-defined successive transformations, but also discusses how these transformations are discoverable from language data, consistent with the fundamental tenets of structural linguistics (which we believe very relevant to LLMs.)
>
> There is no question that the scale of this happening in language is very different from the scale of the three moons problem. However the nature of the phenomenon is the same: when operating in the distant tail of the training distribution, far from any training example, the model can only rely on mathematical structures that were discovered in the body of the training distribution(s) and remain valid in the distant tail.  This is the first time we have model able to discover such structures without cheating (that is without revealing bias introduced by the model designer.)

---

> > ### Comment · Reviewer_k2DW · 2024-11-20
> > **Response to Authors**
> >
> > I thank the authors for their speedy responses to my comments (which I am sure they grow tired of by now!). With the most recent changes, in particular those that reference the compositional structure of language in previous literature, I am quite happy with the paper in its current form. While I still think the research presented is still covering a little too much for a single paper I believe that the authors have several substantial points to make that would be valuable to the community. For this reason I have updated my recommendation for this paper.

---

### Official Review · Reviewer_R52u · 2024-11-03

**Soundness:** 3
**Presentation:** 2
**Contribution:** 3
**Rating:** 8
**Confidence:** 4

**Summary:**

This submission proposes replacing parts of the Transformer block with associative memories. The notion of predictive disentanglement is discussed. The proposed architecture is evaluated on 3-moon problem and small scale language modeling tasks.

**Strengths:**

In general, this is a though-provoking paper with some interesting ideas. Exploration of the relationship between associative memories, attention, and Transformer blocks is valuable, although the current presentation is heavily biased, and omits related ideas from prior work.

The empirical results on language modeling are encouraging, although small scale.

**Weaknesses:**

One problem with this submission is that the presentation almost entirely ignores the work on modern Hopfield networks and dense associative memories, which tackles closely related motivation and ideas. Specifically, the authors’ proposal is closely related to [Energy Transformer (NeurIPS 2024)](https://proceedings.neurips.cc/paper_files/paper/2023/file/57a9b97477b67936298489e3c1417b0a-Paper-Conference.pdf) and related literature, which replaces elements of Transformer block with associative memories. The only cited paper representing that line of work (Ramsauer et al, 2020) is only briefly mentioned. Please notice, that paper only established a connection with associative memory at a single step update, which is different from the setting used in this submission - dynamics unfolded in time with shared weights, if I understood equations 4 and 5 correctly. In contrast, the Energy Transformer formalizes attention operation and the entire Transformer block as a form of associative memory, with recurrent dynamics unfolded in time and shared weights. There are, of course, many differences between these prior works and the current proposal. For instance, it is unclear to me if energy-based description is important at all for this submission. Regardless these similarities/differences this and related works need to be referenced at a prominent place in the introduction and similarities/differences with the current proposal need to be extensively discussed in the revised manuscript.

I have read this submission several times, but to be frank with the authors I am not sure I understand what they mean by predictive disentanglement, which seems to be the core concept here, but defined only vaguely. Unfortunately, I do not have a specific recommendation how one could improve the presentation in section 3. But, at present, it is insufficient for me. What architectural aspects of memory mosaics suggest that predictive disentanglement should hold?

P-mem network is insufficiently explained. Please include specific operations that are used in it in the revised paper (with formulas).

**Questions:**

1. Please highlight a clear definition of predictive disentanglement and explain intuition behind it.
2. Please make the discussion of associative memory based modifications of Transformers more scientifically balanced. At present, the submission leaves a strange impression that the work from FAIR is unreasonably promoted (e.g., Bietti et al, Weston et al., etc), but the work from elsewhere is ignored.
3. Are the proposed networks energy-based or not?
4. Associative memories with Gaussian kernel smoothing (equation 2) have been studied in [End-to-end Differentiable Clustering with Associative Memories, ICML 2023](https://proceedings.mlr.press/v202/saha23a/saha23a.pdf). Please explain in the revised paper how your approach is different/similar to that prior work.
5. What aspect of P-mem makes it persistent? Please include a detailed description of this network with explicit formulas describing how it is implemented. The figure presented in (Fig 6 right) is insufficient for understanding how it is defined.
6. Formula 5 confuses me. Wouldn’t such a definition of values trivially leak the information about the future to the output of the model? In other words, what prevents the model from simply copying that future token and output that copy as a next token prediction?
7. Why were the hyperparameters on language modeling tuned for Transformers, but not for the proposed model?


I am keeping an open mind, and willing to consider increasing the scores depending on authors’ responses.

---

> ### Author Response · Authors · 2024-11-15
>
> Please check our general response with respect to the work on the modern Hopfield networks. Although we find this work direction interesting, we also believe that it is only weakly related and does not provide the right context to understand our paper.
>
> 1) ​​*Please highlight a clear definition of predictive disentanglement and explain intuition behind it.*
>
> Please see our general response 2.
>
> 2) *Please make the discussion of associative memory based modifications of Transformers more scientifically balanced. At present, the submission leaves a strange impression that the work from FAIR is unreasonably promoted (e.g., Bietti et al, Weston et al., etc), but the work from elsewhere is ignored.*
>
> Please see our general response 1 and observe that our use of inference time memories is far more connected to the memory networks of Weston et al (which predate transformers) than to the modern Hopfield network (see our second sentence line 20). We'll make this clear.
>
> 3) *Are the proposed networks energy-based or not?*
>
> They are not.
>
> Remark: Substantial work on associative memories predates Hopfield's seminal work on memories defined through an energy functions, e.g. Anderson's associative memories (J. R. Anderson, "language memory and thought", 1976), Kohonen associative memories (T Kohonen, "Associative memory: a system-theoretical approach", 1977).
>
>
> 4) *Associative memories with Gaussian kernel smoothing (equation 2) have been studied in End-to-end Differentiable Clustering with Associative Memories, ICML 2023. Please explain in the revised paper how your approach is different/similar to that prior work.*
>
> What we do is a direct application of the Nadaraya-Watson estimator of 1964, also known as kernel regression, applied to the list of stored key/value pairs (not using a distributed representation here.)
>
> 5) *What aspect of P-mem makes it persistent? Please include a detailed description of this network with explicit formulas describing how it is implemented. The figure presented in (Fig 6 right) is insufficient for understanding how it is defined.*
>
> Please see our general response 3.  Persistent memories only play a supporting role in this paper. We call them “persistent” because they store key/value pairs that are determined during training and frozen during inference. In contrast the other memories are cleared at the beginning of each test sequence and are filled during inference.
>
> 6) *Formula 5 confuses me. Wouldn’t such a definition of values trivially leak the information about the future to the output of the model? In other words, what prevents the model from simply copying that future token and output that copy as a next token prediction?*
>
> As rephrased in our general response 2, the output of the memory unit $y_T$ at time $T$ only depends on the past of the input sequence. It does not depend on $v_T$ which is only computed at time $T+1$ in order to store a new pair $(k_T,v_T)$ into the memory. There is no leak of information here. You can also see this in equation (7) where the summation stops at $T-1$ in order to exclude $v_T$.
>
> 7) *Why were the hyperparameters on language modeling tuned for Transformers, but not for the proposed model?*
>
> We did this in order to be absolutely certain that our model matches or outperforms the transformer. It is far too easy to obtain good numbers with a subtly fancier hyper parameter search. Therefore we follow a protocol that totally avoids this risk.

---

> > ### Comment · Reviewer_R52u · 2024-11-22
> > **scores updated**
> >
> > Thank you for implementing some of my suggestions in the revised manuscript, and for clarifications. My main concern about absence of discussion regarding the energy-based associative memory modifications of Transformers has been somewhat rectified in the revised version of the manuscript. Clarifications on the architectural proposal are also helpful and appreciated. I have updated my scores.

---

> > > ### Author Response · Authors · 2024-11-22
> > >
> > > Many thanks for pointing out in which ways we could improve our paper, for reading our revised version, and for accepting to reevaluate your score in this improved light.

---

### Official Review · Reviewer_L6NA · 2024-11-05

**Soundness:** 2
**Presentation:** 2
**Contribution:** 2
**Rating:** 3
**Confidence:** 3

**Summary:**

"Memory Mosaics" introduces an architecture where multiple associative memories are employed in unison to enhance predictive capabilities for tasks like language modeling. This architecture aims to combine the benefits of transformer models with greater transparency and efficiency through what the authors term "predictive disentanglement." The paper is technically rich, articulating a clear hypothesis, describing the underlying mechanisms in detail, and providing comparative analysis against existing models.

**Strengths:**

1. The integration of associative memories to replicate and surpass the capabilities of transformers.
2. The concept of predictive disentanglement is novel and also rooted in a solid theoretical framework.
3. The theoretical motivations behind predictive disentanglement are well-explained.

**Weaknesses:**

1. The paper does not provide exhaustive details on the architecture's configuration.
2. Lack of detailed discussion on the choice and impact of hyperparameters.
3.The experimental validation is limited to certain language tasks.
4. Lack of Ablation Studies

**Questions:**

1. Could you dive a bit deeper into the architecture of the Memory Mosaics for us? I'm particularly curious about the specifics of the associative memory units, what kind of activation functions you're using, and how the layers are configured. And also the comparison with other non-transformer models like rwkv and mamba?
2. I'd love to understand more about how you set up the associative memories initially. How do you initialize and update the parameters during training?
3. What led you to choose the hyperparameters that you did for this study? And how does changing these hyperparameters affect the model's performance?
4. Did you perform any hyperparameter tuning experiments? If yes, could you share how you went about it and what the outcomes were?
5. Are there any ablation studies that you've conducted or plan to conduct? It would be really helpful to see how each component of the model contributes to its overall performance.
6. I'm interested in the predictive disentanglement feature of your model. Could you explain how altering or removing this feature affects the model's performance?
7. Have you conducted any scalability tests, particularly in comparison with transformers? I'm curious about how the model scales with larger datasets and how computationally efficient it is.

---

> ### Author Response · Authors · 2024-11-15
>
> 1) *Could you dive a bit deeper into the architecture of the Memory Mosaics for us? I'm particularly curious about the specifics of the associative memory units, what kind of activation functions you're using, and how the layers are configured. And also the comparison with other non-transformer models like rwkv and mamba?*
>
> Please check our general responses as well as sections 2 and 6 in the paper. In particular there are no activation functions in our networks. The only non-linearities are the RMS normalizations and those of equation (7). The layer configurations are explicit and footnote 5 precisely compares the number of weights in our model with those of the GPT-2 model.
>
> 2) *I'd love to understand more about how you set up the associative memories initially. How do you initialize and update the parameters during training?*
>
> Please check our general response 2. The memories operate at inference time. They’re cleared at the beginning of each sequence and acquire new key/value pairs at each time step. The gradient training process only determines the nature of the stored keys and values.
>
> 3) *What led you to choose the hyperparameters that you did for this study? And how does changing these hyperparameters affect the model's performance?*
> 4) *Did you perform any hyperparameter tuning experiments? If yes, could you share how you went about it and what the outcomes were?*
>
> In section 6, we chose to stay as close as possible to the GPT2 architecture, and accordingly chose to reuse the hyper-parameters that were optimal for the GPT2 baseline, without searching for better ones. Although this places Memory Mosaics at a slight disadvantage, it ensures that our positive results are not a consequence of a fancier hyper-parameter search.
>
> 5) *Are there any ablation studies that you've conducted or plan to conduct? It would be really helpful to see how each component of the model contributes to its overall performance.*
>
> There is not much to ablate without making the system incoherent. We have experimented with various replacements for the key and value extraction functions (equation 6) and found that they made little difference as long as they each involved at least two successive inputs.
>
> 6) *I'm interested in the predictive disentanglement feature of your model. Could you explain how altering or removing this feature affects the model's performance?*
>
> Predictive disentanglement is not a feature that we can add or remove at will, but a phenomenon that can be understood when one realizes that the training process is in fact a meta-training process. Please refer to the general response 2.
>
> 7) *Have you conducted any scalability tests, particularly in comparison with transformers? I'm curious about how the model scales with larger datasets and how computationally efficient it is.*
>
> Since writing the paper, we have scaled the architecture to models with a couple billion parameters without particular problems. We also made further strides in leveraging the good properties suggested in figures 8 and 9. However this new research is not ready for publication at the current time.

---

> ### Author Response · Authors · 2024-11-22
>
> Dear reviewer L6NA
>
> Thank you for taking the time to review our paper. We have carefully considered your comments and have provided detailed responses to each of your points. We hope that our answers adequately address your concerns and provide a clearer understanding of our work.
>
> If you have any further questions or concerns, please do not hesitate to leave a comment.

---

> > ### Author Response · Authors · 2024-12-02
> >
> > Dear Reviewer L6NA,
> > have you had time to mull over our rebuttal? Please let us know if you have any remaining questions, we would appreciate a last opportunity to address your concerns.

---

### Author Response · Authors · 2024-11-15
**General response 1 : About existing work on transformer alternatives.**

We thank the reviewer for pointing out that it is useful to explain how our work differs from current work on transformer alternatives.
We'll include a section that will make the following clear.

1)  The computational cost of self-attention grows quadratically with the sequence length. A number of recent papers [CITATIONS] propose transformer alternatives with linear complexity instead.  Our work is orthogonal to that line of research. As should be clear in equation (7), our models have quadratic complexity. We mention, in the last paragraph of the conclusion, that memory mosaics provide opportunities for future work in this direction (lines 520-524), but this neither is the main thrust of this paper nor is its main motivation.

2) Another series of recent papers [citations] share our interest in associative memories but focus on the Hopfield style of associative memories which are governed by an energy function. Our associative memories simply store key/value pairs in the computer memory and use kernel regression to retrieve an interpolated value when provided with an approximate key. This is explained lines 40-47. The argument that justifies such memories is not an energy argument, but a well known statistical argument about kernel regression (see footnote 1. See also the Nadaraya-Watson estimator which is worth citing there.)  Other than relying on associative memories in a broad sense, we are not aware of a precise connection between our approach and the transformer alternatives based on modern Hopfield ideas.

Our contribution is much more closely related to the 2019 paper of Bengio et al (https://arxiv.org/abs/1901.10912) which discusses how a meta-learning objective that favors fast adaptation can disentangle causal factors. The is happening for free in memory mosaics (and this is what predictive disentanglement is about.)

---

> ### Author Response · Authors · 2024-11-15
> **General Response 2 - About associative memories and predictive disentanglement**
>
> The following is an attempt to concisely rephrase the main idea (because the details matter).
>
> In a memory mosaic, each associative memories *operates independently at inference time*, starting empty at the beginning of each token sequence. At each time step $T$, each memory receives a key vector $k_T$ computed from the current and recent tokens. Each memory then interpolates a response $y_T$ on the basis of the previously stored key/value pairs. One time step later, it peeks into the future and computes the value $v_T$ (which contains a bit of information about the next token $x_{T+1}$) and stores the pair $(k_T,v_T)$ into the memory. This is explained lines 98-102, and this is formalized in equation (7), together with a bullet list of points that distinguish this process from the classical self-attention (lines 135-150).
>
> Although the value $v_T$ depends on the future token $x_{T+1}$, the output $y_T$ does not depend on $v_T$ but merely leverages the previously stored key/value pairs to estimate $v_T$.  Therefore there is no leak of future information.  Each memory is simply used as a machine that predicts a bit of future information (described by $v_T$) on the basis of recent information (described by $k_T$) and previously stored key/values pairs.
>
> Importantly, the gradient learning algorithm determines which future bit of information is predicted by each memory (through the parameters that control the computation of the values $v_T$) and which kernels are used to perform the predictions (through the parameters of that control the computation of the keys $k_T$). However the *relation between keys and predicted values is entirely determined at inference time through the memorization of key/values pairs specific to each sequence*. This is explained lines 135-139. This is why we write, lines 155-157, that the gradient training procedure is in fact a meta-learning process whose main purpose is to determine what the memories model at inference time and how they do it.
>
> Through a process that we call *predictive disentanglement*, this meta-learning interpretation reveals an interesting consequence: the gradient training algorithm tends to split the overall prediction task (e.g. predicting the next token in a sentence) into disentangled prediction subtasks assigned to each associative memory unit.  The steamroller figure (Figure 2) summarizes the intuition for this (see also lines 176-212) : the training cost is reduced when the memories are able to provide good predictions $y_T$ of their targets $v_T$ after seeing as few tokens as possible. This is best achieved when the prediction subtasks assigned to each of the memories represent disentangled components of the overall prediction problem, (an idea closely related to https://arxiv.org/abs/1901.10912).
>
> Although similar phenomenon might happen in regular transformers, the memory mosaic perspective makes it considerably easier to understand (yet not fully obvious, unfortunately). Not only this observation is essential to understand why transformer-like architectures are able to learn models with compositional properties, but we also show that, contrary to common beliefs, this is not a consequence of scale since we are able to demonstrate it in a network with only 54 parameters.
>
> If the reviewers find this rephrasing helpful, we'll modify the paper accordingly.

---

> ### Author Response · Authors · 2024-11-15
> **General response 3 - About persistent memories**
>
> Persistent memories were introduced by Sukhbaatar in (2019) as a possible replacement for the fully connected layers features in classical transformer architectures. In our work, persistent memories compute their outputs exactly like the contextual memories (equation 7). The only difference is that the stored key/value pairs do not change at inference time. They have been determined by gradient during training, just like the weights of a fully connected layer. This is explained lines 324-328.
>
> Although replacing the persistent memory layers by fully connected layers of equivalent size does not change the performance of the system, we find interesting that we can train a GPT2-equivalent system that only relies on memories (either contextual memories that store contents pertaining to the current test sequence, or persistent memories whose content have been determined at training time and persists across test sequences). The only non-linearities in the resulting system are those present in equation (7) and in the RMS Normalization steps.
>
> Persistent memories only play a supporting role in our construction. We describe them because we found that they give a conceptually more interesting system without performance penalty.

---

### Author Response · Authors · 2024-11-20
**Revision of the paper**

We just posted a substantial revision of the paper with the intention to account for the comment of the reviewers.

Changes include:
* A new "related work" section listing the lines of work suggested by the reviewers (and the corresponding papers of course) and explaining how our work differs. We also comment on the paper of Bengio et al (2019) which presents ideas closely connected to predictive disentanglement and might help the readers.
* A rework of the paragraphs explaining the detailed operation of the associative memory units and presenting predictive disentanglement, along the lines discussed in the rebuttals.
* A new paragraph at the end of the three moon sections explaining how the moon disentanglement and recombinations is similar to phenomena that one can experience with modern LLMs (albeit on a different scale). We also cite (Harris 1968) to stress out that the compositional structure of language can be described as a strictly mathematical property.
* An additional section in the appendix augments the results presented in figure 7 and provides a substantially larger discussion, too large to fit in the main text.

All changes are shown in blue in the revised pdf (which is accessible using the usual pdf link on open review.)

---

### Meta-Review · Area_Chair_L5U6 · 2024-12-20

**Metareview:**

This paper proposes a neural network architecture that composes of associative memory units. Compared to transformers, it achieves a new capability, called predictive disentanglement, under a meta-learning interpretation of the training process. The paper shows better interpretability and in-context learning performance on language modelling benchmarks.

Pros:
- Novel use of associative memories with a strong theoretical motivation.
- A thought-provoking paper with interesting ideas.
- Encouraging empirical results, showcasing competitive performance.

Cons:
- Insufficient discussion of related works. (addressed in the rebuttal)
- Limited evaluation on relatively small networks.

**Additional Comments On Reviewer Discussion:**

There are extensive discussion between reviewers and authors in the rebuttal process. Reviewers have some common concerns including
- its relationship and novelty compared to prior works on associative memory networks, linear transformers
- understanding the definition of "predictive disentanglement"
- algorithm details
- limited evaluation
Most of the concerns have been addressed after the rebuttal process, and three out of four reviewers raise their ratings in favor of acceptance.

The remaining reviewer has concerns over (1) lack of detailed architecture description (2) lack of disucssion on the choice of hyperparameters and ablation (3) experiments limitation on the scales. Although the reviewer did not engage in the following discussion, the authors detailed response should have clarified the first two points. For the last point on the lack of larger scale experiments, it is a limitation of the current submission. However, given the quality, potential impact of the proposed architecture and thorough evaluation at its current scale, the lack of scalability should not be hold against the acceptance of this submission.

---

### Decision · Program_Chairs · 2025-01-22

Accept (Poster)